# Learning Splitting Heuristics in Divide-and-Conquer SAT Solvers with Reinforcement Learning

**Shumao Zhai**[1]**, Ning Ge**[1,2] *
[1]Beihang University   [2]State Key Laboratory of Complex & Critical Software Environment (CCSE)

## Abstract

We propose RDC-SAT, a novel approach to optimize splitting heuristics in Divide-and-Conquer SAT solvers using deep reinforcement learning. Our method dynamically extracts features from the current solving state whenever a split is required. These features, such as learned clauses, variable activity scores, and clause LBD (Literal Block Distance) values, are represented as a graph. A GNN integrated with an Actor-Critic model processes this graph to determine the optimal split variable. Unlike traditional linear state transitions characterized by Markov processes, divide-and-conquer challenges involve tree-like state transitions. To address this, we developed a reinforcement learning environment based on the Painless framework that efficiently handles these transitions. Additionally, we designed different discounted reward functions for satisfiable and unsatisfiable SAT problems, capable of handling tree-like state transitions. We trained our model using the Decentralized Proximal Policy Optimization (DPPO) algorithm on phase transition random 3-SAT problems and implemented the RDC-SAT solver, which operates in both GPU-accelerated and non-GPU modes. Evaluations show that RDC-SAT significantly improves the performance of D&C solvers on phase transition random 3-SAT datasets and generalizes well to the SAT Competition 2023 dataset, substantially outperforming traditional splitting heuristics.

## 1 Introduction

The propositional satisfiability problem (SAT) is one of the foundational problems in theoretical computer science. It involves determining the assignment of variables that satisfy a given Boolean formula. Due to its NP-complete nature, SAT plays a crucial role in theoretical computer science and has a wide range of applications, including software verification, hardware design verification, combinatorial optimization, and artificial intelligence, etc. Most modern complete SAT solvers primarily employ the conflict-driven clause learning (CDCL) algorithm (Marques-Silva et al., 2021), which has transformed the field by introducing clause learning mechanisms that effectively optimize the search space. CDCL solvers avoid revisiting previously explored paths and concentrate on promising areas of the search space, establishing them as the leading method for tackling complex and challenging SAT problems.

In the realm of modern SAT solvers, parallelism has been embraced to fully harness the computational power of many-core machines (Hamadi & Sais, 2018). Two dominant parallelization strategies have emerged: Divide-and-Conquer (D&C) and Portfolio. The D&C strategy, often employing the guiding path technique, dynamically divides the problem into smaller, more manageable subspaces. Each subspace is independently tackled by individual sequential solvers, allowing the problem to be processed more efficiently in parallel. This method leverages problem structure to enable targeted sub-formula exploration, reducing overall solving time. The Portfolio approach adopts a competitive strategy in which multiple sequential solvers concurrently tackle the entire problem. This competition continues until one solver successfully solves the problem. Strategic diversity among solvers enhances solution coverage and success probability. Although D&C approaches seem to be the natural way to work in parallel, empirical results from annual SAT competitions indicate that Portfolio-based

---

*Corresponding author: Ning Ge (gening@buaa.edu.cn)

solvers currently lead in performance. However, D&C solvers still possess unique advantages in certain domains, such as better scalability in environments with large parallel computing power, like cloud platforms (Heisinger et al., 2020), and have been successfully employed to solve large-scale mathematical problems (Heule et al., 2016).

In modern D&C SAT solvers, the problem is dynamically divided into smaller, more manageable subproblems. The splitting process entails selecting a variable and dividing the problem into sub-formulas by setting the variable to either True or False, which are subsequently solved using CDCL sequential solvers. This partitioning is guided by splitting heuristics, which largely determine the performance of D&C SAT solvers. An effective splitting heuristic minimizes total solving time while ensuring balanced subspaces for optimal load distribution. Splitting heuristics generally fall into two categories: look-ahead and look-back techniques. Look-ahead heuristics evaluate the potential impact of variables to minimize runtime, as demonstrated in the Cube-and-Conquer solver (Heule et al., 2011; Fleury & Heisinger, 2020), which creates smaller subspaces by choosing variables that maximize unit propagations. In contrast, look-back heuristics evaluate variables based on their historical effectiveness in the search, selecting split variables through methods like VSIDS-based variable activity (Audemard et al., 2016), the number of flips of variables (Le Frioux et al., 2019), and propagation rates (Nejati et al., 2017) of variables.

Machine learning methods have already been widely applied to SAT problems (Holden et al., 2021; Guo et al., 2023; Yolcu & Póczos, 2019; Zhang et al., 2020). In recent years, GNNs and RL have shown potential in optimizing branch splitting, phase selection, clause deletion, etc., heuristics in CDCL solvers (Jaszczur et al., 2020; Kurin et al., 2020; Selsam & Bjørner, 2019; Cameron et al., 2020; Han, 2020; Wang et al., 2023; Shi et al., 2023; Liu et al., 2024). For example, Graph-Q-Sat (Kurin et al., 2019; 2020) uses GNNs and RL to optimize branch heuristics in CDCL solvers, significantly reducing the number of iterations. However, Graph-Q-Sat did not effectively reduce overall runtime due to CDCL's requirement for high-frequency variable selection calls. In contrast, splitting heuristics in D&C solvers are equally critical but invoked far less frequently, making RL and GNN-based optimization of these heuristics particularly promising. A more detailed discussion of related work is provided in Appendix A.

In this paper, we present a technique that employs deep reinforcement learning to optimize the splitting euristics within parallel D&C SAT solvers, resulting in the development of the Reinforced Divide-and-Conquer SAT solver (RDC-SAT). When a split is required, the D&C solver parses information from the sequential solver, including learned clauses, variable activities, and clause LBD(Literal Block Distance), and represents this information as a graph. A GNN is then used to evaluate the importance of each variable and select the optimal split variable.

To optimize this model using reinforcement learning, we developed an RL environment within the Painless (PArallel INstantiabLE Sat Solver) framework (Le Frioux et al., 2017; 2019) that efficiently manages the tree-like state transitions characteristic of D&C methods. Additionally, we designed distinct discounted reward functions for satisfiable and unsatisfiable SAT problems. We then trained a model optimized for splitting heuristics using the Distributed Proximal Policy Optimization(DPPO)(Heess et al., 2017) algorithm, resulting in the creation of RDC-SAT. Moreover, considering the computational overhead of the neural network in the SAT solver, we also implemented an approach similar to NeuroBack (Wang et al., 2023), which supports invoking the neural network only once at the initial stage to guide subsequent splits, making it suitable for GPU resource-poor computing environments. To validate our approach, we trained the model on a random dataset at the phase transition and evaluated its effectiveness and generalization on larger-scale problems. Our empirical results demonstrate that RDC-SAT outperforms baseline methods. On the random 3-SAT dataset, RDC-SAT achieved a 25.08% reduction in PAR2 score; on the SAT Competition 2023 dataset—which includes small to medium-sized SAT problems with hundreds of thousands of variables and clauses—it achieved a 14.95% reduction. Even without GPU acceleration, the insights from the initial neural network call effectively guide subsequent splits, ensuring efficient problem-solving with minimal reliance on neural network computations.

## 2   PRELIMINARIES

**SAT problem.** The Boolean Satisfiability Problem (SAT)(Biere et al., 2009) concerns the satisfiability determination of propositional logic formulas. A propositional logic formula is composed

of variables and operators such as AND (conjunction, $\wedge$), OR (disjunction, $\vee$), NOT (negation, $\neg$), and parentheses, typically expressed in Conjunctive Normal Form (CNF). In CNF, a formula is a conjunction of clauses, where a clause is a disjunction of literals, and a literal is either a variable (positive literal) or its negation (negative literal). For example, $(x \vee \neg y) \wedge (\neg x \vee z)$ is a CNF formula, where $x$ and $z$ are positive literals of their respective variables, and $\neg y$ is a negative literal. In the SAT problem, each variable can be assigned one of two values: True or False. If there is at least one assignment of variables that makes the entire formula True, then the formula is deemed satisfiable (SAT); otherwise, it is considered unsatisfiable (UNSAT). During the process of solving SAT problems, CDCL solvers generate numerous learned clauses to aid in resolving the problem efficiently. These learned clauses help reduce the search space by preventing the solver from revisiting conflicting assignments, thus facilitating quicker convergence towards a solution or proving unsatisfiability.

**Divide-and-Conquer SAT solvers.** Modern D&C SAT solvers follow a hierarchical master-slave architecture.(Le Frioux et al., 2019) The master node manages the partitioning of the search space into smaller subspaces, which are tackled independently and in parallel by multiple sequential CDCL solvers. These subproblems are solved concurrently by worker nodes, which share learned clauses with each other to accelerate the solving process and reduce redundant computations. To ensure dynamic load balancing, D&C solvers utilize a work-stealing mechanism. When a worker node becomes idle, the master node identifies an ongoing subproblem currently being processed by another node, splits it into smaller tasks, and assigns one to the idle worker. This dynamic redistribution of tasks enhances computational efficiency. Incremental solvers further optimize this approach by enabling solvers to continue from previous states with retained learned clauses and variable activity scores, minimizing redundancy and improving overall solving speed.

**Graph Neural Networks.** Graph Neural Networks (GNNs) (Wu et al., 2020; Zhou et al., 2020) are designed to handle data structured as graphs, allowing for the modeling of relationships and interactions within data. The GraphNets framework (Battaglia et al., 2018) utilizes an Encoder-Processor-Decoder architecture to effectively process graph-structured data. The encoder maps node and edge features into high-dimensional latent representations. The processor iteratively updates these representations through a message-passing mechanism, where each node aggregates information from its neighbors to refine its embedding. This mechanism enables the network to propagate and integrate information across the graph. Finally, the decoder reconstructs the desired outputs from the node embeddings for tasks such as node classification, graph classification, or link prediction, utilizing the embeddings either directly or through aggregation for graph-level predictions.

**Reinforcement Learning.** Reinforcement Learning (RL) is a machine learning paradigm where an agent learns to make decisions by interacting with its environment, maximizing cumulative rewards. Central to RL is the classic Markov Decision Process (MDP), which models the decision-making problem through states $s \in S$, actions $a \in A$, state-transition probabilities $P(s'|s, a)$, and a reward function $R(s, a)$. This framework assumes the state transitions are generally linear, meaning the next state $s'$ depends only on the current state $s$ and action $a$. Advanced RL algorithms like Actor-Critic methods use a policy network (actor) to select actions and a value network (critic) to estimate the value function, improving both networks iteratively. Proximal Policy Optimization (PPO) (Schulman et al., 2017) introduces a clipped surrogate objective function to stabilize policy updates, preventing drastic changes and ensuring more reliable learning. Distributed Proximal Policy Optimization (DPPO)Heess et al. (2017) extends PPO by employing multiple agents across distributed environments, leveraging data parallelism to speed up learning while maintaining policy stability.

## 3 INTERACTIVE DIVIDE-AND-CONQUER ENVIRONMENT

We selected Painless (Le Frioux et al., 2017; 2019) as the foundational parallel divide-and-conquer framework. The Painless framework is a modular and high-performance parallel SAT-solving system, ideal for comparing different heuristics. It uses a master-slave architecture with a work-stealing strategy for dynamic load balancing and supports clause sharing to enhance solving efficiency. For our experiments, we used the best-performing settings identified, including MapleCOMSPS (Liang et al., 2016b;a) as the sequential solver and the "alltoall" clause-sharing strategy. Building on this foundation, to construct a reinforcement learning environment suitable for optimizing splitting heuristics and handling tree-structured state transitions, we implemented features to extract SAT problem states from the SAT solver, store various information required for training using a binary

tree, and retain only simple interfaces. To use Python for model training, we encapsulated the divide-and-conquer environment as a Python library using Pybind11 and created the *PainlessEnv* environment in Python. Specific implementation details can be found in Appendix B.

## 3.1 ENVIRONMENT INTERFACE

Our custom *PainlessEnv* environment is specifically designed for the needs of divide-and-conquer SAT problems, addressing the complexities of parallel and branching operations in SAT solving. It manages a binary tree structure to support the non-linear state transitions characteristic of these problems, unlike the linear transitions typical in traditional environments like Gym (Brockman et al., 2016). *PainlessEnv* primarily provides two interfaces for agent interaction: 'reset' and 'step'. The reset method initializes a new SAT problem and provides the initial state of the first subproblem that needs to be partitioned. Unlike Gym's 'step' method, which returns the next observation, reward, and done flag, our 'step' method only returns the next subproblem state and a 'done' flag. This is because, in the tree-structured state transitions, the next state is not always a direct successor of the previous state, and the reward can only be accurately determined from the tree structure. Therefore, our 'step' method does not return the reward directly.

## 3.2 MANAGING STATE TRANSITIONS WITH BINARY TREES

Due to the non-linear state transitions inherent in D&C problems, *PainlessEnv* cannot produce a continuous sequence of states and rewards for training. Instead, it utilizes a binary tree, as illustrated in Figure 1, to store all critical information during the solving process. The root node denotes the initial problem, internal nodes correspond to subproblems requiring splitting, and leaf nodes signify fully solved subproblems. When a worker requests a division or completes a solution, a new node is added to the tree with all pertinent data. After solving a SAT problem, the reinforcement learning agent retrieves training information from this binary tree. Additionally, In a D&C solver for SAT problems, once a subproblem is proven satisfiable, it indicates that the entire SAT problem is satisfiable. Typically, the D&C solver would interrupt all other active workers and return the result. However, for reinforcement learning training, the other unresolved subproblems are still meaningful. Interrupting the process leads to missing leaf nodes in the binary tree, which is suboptimal for training. To address this, even if a subproblem is confirmed as satisfiable, we continue processing until all subproblems are resolved before returning a result, ensuring a complete and informative dataset for training.

## 3.3 SAT SUBPROBLEM STATE EXTRACTION

Each time a split is required, the Painless environment interrupts a sequential solver and extracts basic information about the SAT problem to represent its state. MapleCOMSPS categorizes learned clauses into three levels: core, tier2, and local. Core learned clauses are considered the most crucial. They are generally derived from conflicts that involve decisions at deeper levels of the decision tree that are usually kept permanently. Tier2 learned clauses are of intermediate importance. They are often derived from conflicts at shallower depths and are kept as long as they are deemed useful. Local learned clauses are less critical and more temporary, typically generated from recent conflicts and are discarded periodically to manage memory and performance overhead. Additionally, it maintains variable activity values used in branch heuristics, the LBD values of clauses to evaluate their utility, and other statistical information such as the number of decisions and unit propagations. We extract these learned clauses at different levels, along with variable activity scores and other pertinent data, to comprehensively represent the current solving state of the SAT problem. This information is then represented as a graph, which our environment can be configured to utilize.

## 4 REINFORCED DIVIDE-AND-CONQUER SAT

We utilized a state representation and a GNN similar to Graph-Q-Sat (Kurin et al., 2020), with modifications to incorporate additional information that better represents the evolving state of the SAT problem. Additionally, we replaced DQN with an Actor-Critic network and trained it using DPPO.

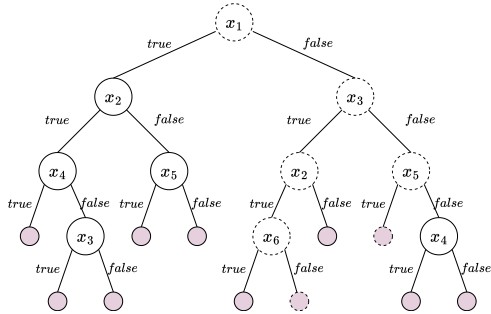
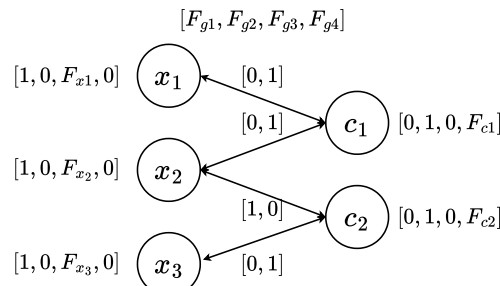

Figure 1: Binary tree maintained by *PainlessEnv* to store process information. Each node in the tree stores problem states, splitting times, solving times, split variables, satisfiability status, and other relevant information. Internal nodes represent subproblems that have been split, while leaf nodes represent solved subproblems. Dashed lines indicate satisfiable subproblems.

Figure 2: State representation of Boolean formula $(x_1 \lor x_2) \land (\neg x_2 \lor x_3)$. The first position of the node feature indicates node type: 1 for variable nodes and 0 for clause nodes, and vice versa for the second position. $F_x$, $F_c$, and $F_g$ represent variable activity features, clause LBD features, and global SAT problem statistics, respectively. Edge features represent the polarity of literals.

### 4.1 STATE REPRESENTATION

In RDC-SAT, the SAT problem is structured as a graph shown in Figure 2 where nodes represent variables and clauses, with node and edge features indicating variable polarity and type. We extract detailed information from the Maple solver, including variable data and learned clauses at various levels. Incorporating learned clauses provides a more dynamic reflection of the SAT problem's state. Variable activity levels and clause LBD values are normalized to enhance message passing efficiency. Additionally, global characteristics like restart counts, decision counts, conflict counts, and unit propagation counts are included as global features. This comprehensive representation captures the dynamic state changes in the SAT problem, enabling the GNN to process node and edge data more effectively.

### 4.2 REWARD FUNCTION FOR TREE-STRUCTURED STATE TRANSITIONS

In traditional reinforcement learning, rewards are typically calculated for linear state transitions, where the reward for each state is computed from the final state backward through the sequence, applying a discount factor at each step. This process involves calculating the reward for the final state first and then propagating it backward through the sequence. Correspondingly, for tree-structured state transitions, calculating the reward for each state requires first computing the discounted rewards for its child states. This necessitates using a post-order traversal to ensure that the discounted rewards of the child states are available before calculating the reward for the current state. Once the discounted rewards of the child states are obtained, they can be integrated into the discounted reward of current state using methods such as summation, averaging, taking the maximum, or the minimum of the child states' rewards. To illustrate this, consider a binary tree, where each state has two child states. Mathematically, if $R_L$ and $R_R$ are the discounted rewards of the left and right child states, and $r_s$ is the immediate reward for the current state, the current state's reward $R$ can be calculated as

$$R = r_s + \gamma \cdot f(R_L, R_R)$$

The primary goal of this optimization is to minimize solving time. Thus, we simply use the negative solving time as the reward, making the maximization of this reward equivalent to minimizing the solving time. After solving SAT problems, we extract the splitting time and solving time from the binary tree in *PainlessEnv* to calculate the actual solving time for each state. For unsatisfiable problems, the total solving time is the sum of the splitting time and the longer solving time of the two subproblems. In contrast, for satisfiable problems, the total solving time is the sum of the splitting time and the shorter solving time of the satisfiable subproblems. The following formulas encapsulate

the calculation of discounted reward:

$$R_U = -\left(T_s + \max\left(\Delta T_{Uc} + \gamma \cdot |R_{Uc}|\right)\right)$$

$$R_S = -\left(T_s + \min\left(\Delta T_{Sc} + \gamma \cdot |R_{Sc}|\right)\right)$$

Here, $R_U$ and $R_S$ are the discounted rewards for UNSAT and SAT problems, respectively; $T_s$ is the splitting time, which is the time taken to analyze the problem state and decide the split variable; $|R_{Uc}|$ and $|R_{Sc}|$ are the absolute values of the discounted rewards for the child subproblems of UNSAT and SAT states, respectively. If a corresponding child state is a leaf node, its discounted reward is zero; $\Delta T_{Uc}$ and $\Delta T_{Sc}$ are the time intervals from the completion of the division to the return of the next child subproblem state in UNSAT and SAT scenarios, respectively; and $\gamma$ is the discount factor applied to future rewards. Typically, $\gamma$ is less than 1 to prioritize immediate rewards over future rewards. However, in our study, we set $\gamma = 1$ because the solving process terminates within a definite time frame. This ensures that each action considers the entire solving process, and the reward directly reflects the total solving time under the current policy and action.

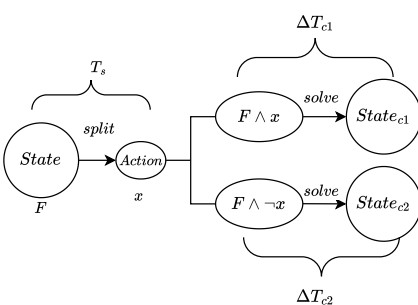

Figure 3: Calculation of discounted reward in tree-structured state transitions.

### 4.3 Neural Network, Action Definition, and Objective Function

We use a Graph Neural Network (GNN) with an Encoder-Processor-Decoder architecture (Battaglia et al., 2018) to extract features, followed by an Actor-Critic network to make decisions. The Encoder transforms initial features into higher-dimensional representations. The Processor integrates local and global information through message passing. The Decoder outputs variable, clause, and global features. The RL action corresponds to selecting a variable as the split variable during the Divide-and-Conquer process. The Actor network assigns probabilities to valid variables, excluding those already chosen as split variables or eliminated during the solving process, ensuring the action space only contains currently selectable variables. The Critic network estimates the value of the current state based on clause and global features. We use the Proximal Policy Optimization (PPO)(Schulman et al., 2017) objective function. This objective combines policy loss, value loss, and entropy loss to optimize the learning process.

### 4.4 Training and Deployment

**Training data.** We utilize randomly generated 3-SAT problems at the phase transition as our training data. The phase transition in SAT problems is characterized by a specific ratio of clauses ($n$) to variables ($m$), given by $n = 4.258m + 58.26m^{-\frac{2}{3}}$ (Crawford & Auton, 1996), where the likelihood of satisfiability sharply transitions from high to low as the number of clauses increases relative to the number of variables. This critical point presents the greatest computational challenge. Although these datasets may initially exhibit a uniform structure, CDCL solvers in D&C solvers can quickly generate a large number of learned clauses, making the subsequent SAT problems input to the neural network inherently rich in structural features. We selected two datasets: one with 300 variables and 1,278 clauses, and another with 350 variables and 1,491 clauses (hereafter referred to as the 300-1278 and 350-1491 datasets, respectively), each comprising 1,000 instances with a balanced ratio of satisfiable to unsatisfiable instances. These training datasets can be solved within seconds to tens of seconds, allowing for the generation of a large amount of training data. All these datasets can be solved within a reasonable timeframe, generating complete binary trees that are used for subsequent extraction of training data.

**Training.** We implemented DPPO using a multi-agent architecture to enhance both exploration and learning efficiency. The system architecture includes five exploration agents continuously solving SAT problems and collecting training data, complemented by a single dedicated training agent. Each exploration agent runs 16 threads to solve SAT problems continuously. Upon completion, these agents parse the binary tree to extract training data, which is then transmitted to the training agent via

a message queue. The training agent systematically retrieves this data from the queue, stores it in a replay buffer, and samples from this buffer to form batches for neural network training. We represent the state using a graph constructed from variables, original clauses, core learned clauses, and tier2 learned clauses, all extracted from the Sequential CDCL solver. We use the Adam optimizer to train the model. Post-training, the updated network parameters are synchronized across all exploration agents to maintain consistency and optimize policy performance. An $\epsilon$-greedy strategy is implemented to ensure a balance between exploration and exploitation, with the epsilon parameter decreasing progressively over time. Additionally, all rewards are normalized during training to maintain a consistent learning scale and enhance algorithmic stability. The exploration agents use only CPU resources, while the training agent uses only GPU resources. For exploration processes, we used LRB as the branch heuristics, as it performed better on the random dataset. Training took approximately 2-3 days. After completing this training, we switched the branch heuristics to VSIDS and trained a new model, which took about half a day. This training operation is hosted on a Linux server equipped with an Intel(R) Xeon(R) Platinum 8373C CPU, featuring 72 physical cores and 144 logical cores, supplemented by 256 GB of memory and a Tesla A100 GPU with 80 GB of memory. During training, we observed that even though random 3-SAT problems at the phase transition initially have just over a thousand clauses, the states used for training showed that the CDCL solver generated approximately ten to fifty thousand core and tier2 learned clauses, ensuring a rich structural representation.

**Deployment.** After the training phase, we utilized TorchScript to export the model and integrate it with the Painless framework, thereby creating the RDC-SAT solver. This solver is a complete D&C SAT solver that no longer depends on Python, and it can dynamically generate variable probability rankings required for splitting decisions using the neural network. The solver can choose to run the neural network's forward propagation on either CPU or GPU. However, the relatively high overhead of neural networks often becomes a bottleneck in SAT solving with GNNs, especially when GPUs are unavailable and the forward propagation is done using CPUs. When multiple threads call the neural network simultaneously, resource contention may occur, slowing down the solving process. To address this and enable RDC-SAT to function effectively without GPU support, we implemented a new strategy: the neural network is used only once during the first required split to generate a ranked list of candidate split variables. For all subsequent splits, the solver directly selects split variables from this precomputed ranking. Additionally, when the problem size is very large, leading to excessive overhead from neural network calls or potential memory limitations, RDC-SAT will switch to the default splitting heuristic.

## 5 EVALUATION

In this chapter, we compare the performance of different splitting heuristics within the Painless framework across various datasets. We evaluate the performance of RDC-SAT against traditional splitting heuristics and analyze the cost of using GNNs.

### 5.1 SETUP

**Solvers.** We used the Painless framework to evaluate the performance of our RDC methods compared to baseline divide-and-conquer solvers. To ensure a fair comparison, we kept all configurations consistent except for the splitting heuristics. Specifically, we used FLIPS (Audemard et al., 2014b), VSIDS(Audemard et al., 2016), and PR (Nejati et al., 2017) splitting strategies from the work (Le Frioux et al., 2019), with P-CLONE-FLIPS showing the best overall performance, outperforming Treengeling and MapleAmpharos. Our RDC-SAT utilized core and tier2 learned clauses to construct the graph, employing two modes: RDC-GPU-always and RDC-CPU-once. RDC-GPU-always uses the GPU to accelerate neural network calls, selecting the split variable with GNN every time it splits. RDC-CPU-once, on the other hand, does not rely on GPU and only uses the CPU to call the GNN and generate variable sorting. For subsequent splitting, the worker uses the variable probability sorting to select split variables. Due to the poor performance of VSIDS on random datasets compared to LRB (as shown in Appendix E.2), we used Maple's LRB branch heuristics for the random dataset, and the RDC used the LRB model. In contrast, for the SAT Competition dataset, we used Maple's VSIDS branch heuristics, and the RDC used the VSIDS model.

**Dataset.** We evaluated RDC-SAT on two datasets. (1) To investigate whether RDC-SAT has effectively learned the splitting heuristics, we randomly selected 200 SAT instances from the random

dataset at the phase transition. This set includes 100 problems from the 450-1917 dataset and 100 from the 500-2129 dataset, with a 1:1 ratio of satisfiable to unsatisfiable instances. This dataset provides a suitable level of difficulty for the baseline solvers, enabling us to distinguish their performance. (2) To evaluate whether RDC-SAT, trained on random datasets, can generalize to more extensive, application-oriented datasets, we screened 305 small to medium-sized SAT problems from the 400 instances in the SAT Competition 2023. These problems, which make up the majority of the competition's instances, have a total number of variables and clauses below one million. The SAT Competition dataset includes a diverse range of difficult SAT instances, such as industrial applications and handmade problems. Since our model was trained solely on random datasets, without any knowledge of application-specific or handmade instances, this dataset serves as an ideal benchmark for evaluating the generalization capability of RDC-SAT.

All experiments were conducted on the same machine used for training, with each solver allocated 16 threads. RDC-CPU-once was also limited to using 16 CPU cores for running the neural network's forward propagation process. The timeout for each instance was set to 5000 seconds. For instances that timed out, we used the same PAR2 score as in the SAT Competition, treating the timeouts as requiring double the timeout duration to solve. To analyze the performance of different splitting heuristics, we recorded the splitting and solving time. In our small-scale tests, we found that the PR method was not well-suited for solving random problems, and thus we did not evaluate PR on the random dataset.

## 5.2 EVALUATION RESULT

Table 1 presents all the results. Figure 4 illustrates the comparison of wall-clock time and problems solved across different splitting heuristics on two datasets. Instances that no solver could successfully solve within the time limit were excluded from the statistics.

Table 1: Comparison of Splitting Heuristics across Datasets. #SAT, #UNSAT, and #ALL indicate the number of SAT problems, UNSAT problems, and total problems solved within the time limit. Clock Time is the average wall-clock time in seconds. PAR2 is the penalized average runtime. Splits are the average number of splits per problem. Splitting Time represents the average CPU time spent by all threads on splitting SAT subproblems. Solving Time represents the average CPU time spent by all threads using the Sequential CDCL solver to solve SAT subproblems. All time values are in seconds.

| Dataset | Splitting Heuristics | #SAT | #UNSAT | #All | Clock Time | PAR2 | Splits | Splitting Time | Solving Time |
|---|---|---|---|---|---|---|---|---|---|
| Random (141) | RDC-GPU-always | **76** | **53** | **129** | **1064.43** | **1489.96** | 60.42 | 20.78 | **16870.24** |
| | RDC-CPU-once | 73 | **53** | 126 | 1238.04 | 1769.95 | 65.84 | 0.624 | 19316.30 |
| | FLIPS | 74 | 50 | 124 | 1385.80 | 1988.64 | 49.21 | 0.00037 | 22090.20 |
| | LRB | 70 | 50 | 120 | 1484.81 | 2229.49 | 74.77 | 0.00054 | 23617.79 |
| SAT COMP 2023 (140) | RDC-GPU-always | 50 | 85 | 135 | **764.00** | **942.57** | 162.19 | 142.47 | **10678.02** |
| | RDC-CPU-once | 50 | 85 | 135 | 882.10 | 1060.67 | 191.95 | 8.77 | 11737.56 |
| | FLIPS | 50 | 83 | 133 | 858.25 | 1108.25 | 80.02 | 0.0146 | 12296.52 |
| | VSIDS | 46 | 84 | 130 | 901.12 | 1258.26 | 108.44 | 0.0013 | 12602.74 |
| | PR | 49 | 72 | 121 | 1269.57 | 1948.14 | 104.19 | 0.0155 | 19232.39 |

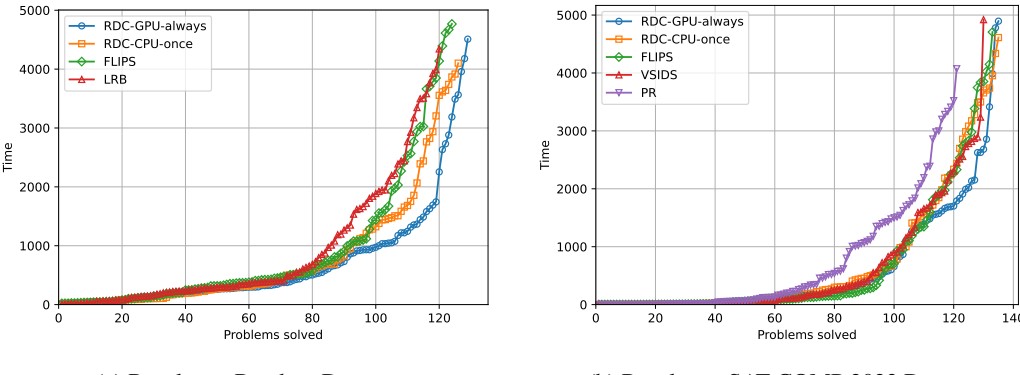

(a) Results on Random Dataset
(b) Results on SAT COMP 2023 Dataset

Figure 4: Comparison of wall-clock time and problems solved across splitting heuristics on two datasets.

**Evaluation on Random 3-SAT Dataset.** For the Random 3-SAT dataset, the RDC-GPU-always and RDC-CPU-once modes were evaluated against traditional heuristics FLIPS and LRB. RDC-GPU-always successfully solved 129 out of 141 problems, while RDC-CPU-once solved 126, both significantly outperforming the traditional splitting heuristics in terms of average clock time and PAR2 score. Specifically, RDC-GPU-always achieved the best performance with an average clock time of 1064.43 seconds and a PAR2 score of 1489.96. Compared to the best baseline method, FLIPS, RDC-GPU-always reduced the PAR2 score by approximately 25.08%, demonstrating the model's capability to effectively learn and apply splitting heuristics from the random dataset, thus significantly enhancing solving efficiency. Even RDC-CPU-once, which only utilizes the initial SAT problem state for its analysis, reduced the PAR2 score by approximately 10.99%, showing substantial performance improvement over traditional heuristics.

**Generalization to SAT Competition Dataset.** The RDC-SAT model, trained solely on random 3-SAT problems at the phase transition without any additional dataset information, was further evaluated on the SAT Competition dataset to test its generalization capability. This dataset includes a diverse range of problems, many of which are application-specific, providing a comprehensive test of RDC-SAT's robustness. Both RDC-GPU-always and RDC-CPU-once demonstrated competitive performance. Specifically, both methods successfully solved 135 problems, with RDC-GPU-always achieving the best average clock time of 764.00 seconds and a PAR2 score of 942.57. Compared to the best traditional method, FLIPS, which had a PAR2 score of 1108.25, RDC-GPU-always reduced the PAR2 score by approximately 14.95%, showcasing its effectiveness across varied problem types. RDC-CPU-once, while solving the same number of problems, recorded slightly higher average clock time and PAR2 score of 1060.67, yet still achieved a PAR2 reduction of approximately 4.29%, indicating substantial performance improvement even with limited reliance on neural network calls. These results underscore the ability of RDC-SAT to generalize well beyond its training conditions.

**Analysis of Splitting Time and Solving Time.** To analyze the solving performance of our methods, we consider two key metrics: Splitting Time, which represents the total splitting time across all threads, and Solving Time, which represents the total solving time across all threads. In SAT solvers, incorporating a GNN is beneficial only if its performance improvements outweigh the overhead. In D&C solvers, the average number of splits per problem is relatively low; for example, in the random dataset, it ranges from 49.21 to 74.77 across various splitting heuristics (see Table 1). Therefore, even if the GNN is invoked at each split, it does not introduce significant performance overhead. Figures 5a and 5b illustrate the relationship between the splitting time per split and the total number of nodes for RDC-GPU-always and RDC-CPU-once on the two datasets. From the figures, it can be observed that the Splitting Time is approximately proportional to the total number of nodes. Notably, in RDC-GPU-always, the average time per split is less than 1 second. In RDC-CPU-once, although the initial GNN call takes slightly longer, subsequent splits are faster because they utilize precomputed variable rankings. When considering total CPU time, the Splitting Time of RDC-GPU-always accounts for only 0.12% and 1.33% of the total Solving Time on the random and SAT Competition datasets, respectively. Similarly, for RDC-CPU-once, it accounts for just 0.003% and 0.075%. Despite the slightly higher Splitting Time compared to baseline methods, the RDC methods significantly reduce the overall Solving Time. Specifically, RDC-GPU-always reduces the Solving Time by approximately 23.63% and 13.17% on the random and SAT Competition datasets, respectively, compared to the best-performing baseline, FLIPS. RDC-CPU-once achieves reductions of approximately 12.56% and 4.54%, respectively. Therefore, although the Splitting Time of both RDC-SAT methods is marginally higher than that of the baseline methods, they substantially decrease the Solving Time, leading to superior overall performance compared to the baselines.

# 6 DISCUSSION

In this paper, we propose a method to optimize splitting heuristics in D&C SAT solvers using reinforcement learning. We built a reinforcement learning environment tailored for Divide-and-Conquer tasks and trained a model, resulting in the RDC-SAT solver. Compared to baseline methods, RDC-SAT achieves significant performance improvements in wall-clock time. To the best of our knowledge, this is the first work addressing tree-structured state transitions induced by Divide-and-Conquer tasks using reinforcement learning. However, our focus was on exploring the potential of reinforcement learning and GNNs in optimizing splitting heuristics, rather than constructing a state-of-the-art (SOTA) parallel D&C SAT solver. Therefore, we did not modify any other components of

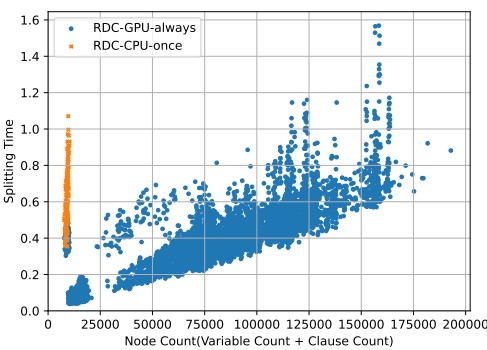
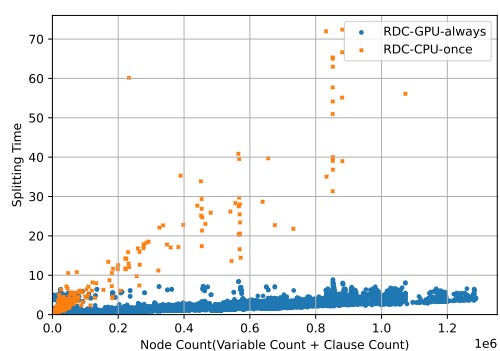

(a) Splitting Time on Random Dataset. RDC-GPU-always: Avg. 72803.28 nodes, 0.344s per split. RDC-CPU-once: Avg. 9043.16 nodes, 0.624s per split.

(b) Splitting Time on SAT Competition 2023 Dataset. RDC-GPU-always: Avg. 248184.79 nodes, 0.878s per split. RDC-CPU-once: Avg. 191098.54 nodes, 8.77s per split.

Figure 5: Analysis of splitting Times for RDC-SAT on two dataset.

the Painless framework or train a general model on large-scale datasets containing multiple types of instances, leaving room for further optimization to reach better performance.

**Integration of GNNs with SAT Solvers.** The integration of GNNs with SAT solving has become a hot research area in recent years. However, for SAT solvers, which are performance-critical tools, the overhead introduced by GNNs—including computational cost and GPU memory usage—is non-negligible, making it challenging to scale to very large SAT problems. Leading SAT solvers have not adopted GNNs. For example, attempts to swap CDCL solvers' branch heuristics with GNNs show that while GNNs can improve decision-making, their significant overhead might extend overall solving time. Our work expands the application of GNNs in SAT solving to a new direction. In a Divide-and-Conquer solver, the splitting heuristic significantly impacts performance, but its invocation frequency is relatively low, allowing RDC-SAT to improve wall-clock time. To extend GNNs to larger-scale SAT problems and reduce overhead, future efforts should prioritize reducing both the size and computational cost of the neural networks. Currently, the GNN employed in RDC-SAT is relatively large, resulting in substantial GPU memory consumption. We plan to reduce the size of the neural network to decrease memory usage and improve efficiency. For large SAT problems, sampling methods can be explored to construct the graph. Techniques such as sampled message passing could further reduce the overhead of GNNs. Minimizing GNN calls is another crucial focus. For example, using the RDC-once method proposed in this work, the GNN is called only initially or at specific points to generate information that guides subsequent decisions. Currently, RDC-once only calls the neural network once initially, without using any new information generated during the solving process. In the future, we will explore a hybrid approach that integrates initial GNN-generated information with default strategies for decision-making. Additionally, training specialized models for domain-specific problems(Li et al., 2022) may yield significant performance improvements for these applications.

**Further Improvements for RDC-SAT.** The development of D&C solvers has lagged behind portfolio-based parallel solvers in recent years. Many D&C solvers, including the Painless framework we used, have not incorporated high-performance sequential solvers like Kissat and CaDiCaL or recent optimizations. RDC-SAT currently uses default configurations of the Painless framework, such as MapleCOMSPS as the sequential solver. Replacing MapleCOMSPS with better-performing solvers, optimizing other components, and exploring different configurations could lead to substantial improvements. Additionally, since our experiments aimed to evaluate RDC-SAT's performance, we trained the model only on the random dataset without training a general model. Training on larger and more diverse datasets could enhance RDC-SAT's effectiveness and generalization. Furthermore, RDC-SAT uses limited information to represent states, focusing on key features of variables and clauses. Future work could explore incorporating additional features to achieve greater performance enhancements.

In future work, we plan to implement these optimizations and evaluate the performance of RDC-SAT on larger datasets to further enhance its effectiveness.

ACKNOWLEDGMENTS

This work was supported by National Key Research and Development Program of China Grant 2022YFB3305102, Natural Science Foundation of China Grant 61902011, and State Key Laboratory of Complex & Critical Software Environment (CCSE-2024ZX-16).

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

# A    RELATED WORK

## A.1    DIVIDE-AND-CONQUER SAT SOLVERS AND THEIR SPLITTING HEURISTICS

Divide-and-Conquer (D&C) SAT solvers have gained prominence due to their ability to leverage parallel computing resources to tackle large and complex SAT problems efficiently. These solvers divide the problem into smaller subproblems, which are then solved independently. A crucial aspect of their performance is the heuristic used to select the best split variable, which significantly impacts the overall solving time and the balance of the resulting subspaces.

Treengeling (Fleury & Heisinger, 2020) and Paracooba (Heisinger et al., 2020) are advanced SAT solvers that employ the Cube-and-Conquer method to harness parallelism. Treengeling utilizes a look-ahead heuristic during the Cube phase to divide the SAT problem into numerous subproblems, or cubes. This heuristic selects the split variable based on the number of unit propagations, aiming to create balanced subspaces. Treengeling generates more cubes than the available computational cores to ensure efficient utilization of all resources. Each cube is then solved independently by sequential CDCL solvers in the Conquer phase. Similarly, Paracooba extends the Cube-and-Conquer method to a distributed environment. It divides the SAT problem into many cubes using a look-ahead heuristic and distributes these cubes to different nodes across a network. Each node runs a CDCL solver to independently solve its assigned subproblems. By aggregating the computational power of multiple machines, Paracooba effectively handles significantly larger SAT problems, ensuring better load balancing and reducing overall solving time.

In contrast to look-ahead techniques, some solvers use look-back methods that dynamically analyze the search performed by the solver, as well as formula statistics, to identify the best candidate at the current splitting point. For example, Ampharos (Audemard et al., 2016) picks the variable with the highest VSIDS activity, while MapleAmpharos (Nejati et al., 2017) uses a propagation-rate (PR) based splitting heuristic. This heuristic ranks variables by the ratio of the propagations they cause to the number of times they are branched on, selecting the highest-ranking variables for splitting. Another explored track is the number of flips of the variables (Audemard et al., 2014b;a). A flip occurs when a variable is propagated to the reverse of its last propagated value. Ranking the variables according to the number of their flips and choosing the highest one as a split variable helps to generate search subspaces with comparable computational time. This approach can also be used to limit the number of variables on which the look-ahead propagation is applied by preselecting a predefined percentage of variables with the highest number of flips. MaplePainlessDC (Nejati et al., 2020) uses a machine learning-based splitting heuristic to optimize divide-and-conquer parallel Boolean SAT solvers. The paper developed pairwise ranking models and minimum ranking models using a random forest classifier, comparing and selecting split variables based on structural features of the formula and dynamic probing statistics.

The Painless Framework is a modular and high-performance parallel SAT solving framework that supports the implementation of various components, including multiple splitting heuristics. It allows developers to create parallel SAT solvers that can compete with state-of-the-art solvers. The framework implements several splitting heuristics, ensuring its solvers can match the performance of leading SAT solvers. Our reinforcement learning environment and the RDC-SAT solver are developed based on the Painless framework. This framework was also used for comparative experiments to evaluate the effectiveness of our approach.

## A.2    OPTIMIZING SAT SOLVING WITH GNNS AND RL

Graph Neural Networks (GNNs) and Reinforcement Learning (RL) have been increasingly applied to optimize SAT solving, showing promising results. Research like Graph-Q-Sat has utilized GNNs to learn branch heuristics in CDCL solvers, replacing VSIDS. Similarly, RL techniques have been used to train heuristics to enhance decision-making in SAT solvers. However, the overhead of calling GNNs for each variable decision in CDCL solvers is very high, making it difficult to apply to real-world problems. Consequently, recent research in this field has focused on reducing the overhead and frequency of GNN calls.

Graph-Q-Sat(Kurin et al., 2020) is an approach that improves the heuristics of SAT solvers by employing GNNs and reinforcement learning. It uses value-based reinforcement learning to train a branching heuristic that significantly reduces the number of iterations compared to VSIDS. However,

it requires calling GNNs at each decision point, which incurs high computational costs and does not reduce the actual wall-clock solving time. To mitigate this issue, other research, such as Neuro-Core (Selsam & Bjørner, 2019), predicts the unsatisfiable core through periodic GNN inferences, reducing the frequency of GNN calls. While this approach enhances solver performance, it still demands substantial computational resources and relies heavily on GPUs. Neuroglue (Han, 2020) enhances SAT solvers by predicting glue variables, which are critical for efficient clause learning. This method periodically updates variable scores using GNN-based insights, significantly improving solver performance without incurring substantial computational costs. NeuroBack (Wang et al., 2023) builds upon these methods by making offline GNN predictions about variable phases before the SAT solving begins, thus eliminating the need for frequent online inferences and reducing computational costs. NeuroBack guides CDCL solvers like Kissat with a single GNN call for phase predictions, increasing performance on recent SAT competition benchmarks. SATformer (Shi et al., 2023) uses a GNN to extract clause representations only once, reducing computational costs, and applies a Transformer to predict unsatisfiability. Integrated into solvers like CaDiCaL and Kissat, SATformer improves solving efficiency over baseline methods. NeuroSelect (Liu et al., 2024) employs a GNN to guide the deletion of learned clauses, making only a single GNN call per instance. On instances with fewer than 400,000 nodes from the SAT Competition 2022, NeuroSelect improves performance over Kissat These methods primarily optimize branch heuristics and phase heuristics in CDCL solvers, but the overhead of GNNs remains a challenge.

In contrast, our approach focuses on the importance of splitting heuristics in Divide-and-Conquer (D&C) solvers. The invocation frequency of these heuristics is relatively low, making it feasible to use GNNs without incurring high computational costs. Our RDC-GPU method leverages GNNs to significantly optimize D&C solver performance with minimal GNN calls. Even without GPUs, we can adopt a strategy similar to NeuroBack. By using RDC-CPU-once, we make a single GNN call to guide subsequent splitting decisions, thereby optimizing splitting heuristics while minimizing GNN overhead.

# B  IMPLEMENTATION DETAILS OF INTERACTIVE DIVIDE-AND-CONQUER ENVIRONMENT

## B.1  DIVIDE-AND-CONQUER IN PAINLESS FRAMEWORK

The Painless framework (Le Frioux et al., 2017; 2019) is a modular and high-performance parallel SAT-solving framework that is both flexible and ideal for comparing different algorithms. Within Painless, the D&C algorithm is implemented using a master-slave architecture that employs a work-stealing strategy to achieve dynamic load balancing. Whenever a CPU core becomes idle, the master thread interrupts an active worker, splits its subproblem, and assigns it to two separate workers. Painless also supports clause sharing, allowing multiple workers to exchange learned clauses to speed up the solving process. This design makes the framework highly adaptable and efficient, empowering researchers to explore and compare various heuristics for parallel SAT solving.

The Painless framework incorporates several D&C solvers, which can compete effectively with the leading solvers available. For the basic configuration in our experiment, we used the best-performing settings identified through experimentation. This includes utilizing MapleCOMSPS (Liang et al., 2016b) as the sequential solver, the "alltoall" clause-sharing strategy, and the "clone" strategy. Specifically, MapleCOMSPS, which won the SAT Competition in 2016, serves as the sequential solver. The "alltoall" clause-sharing strategy ensures that all solvers directly share clauses with an LBD below 4, which helps to improve the efficiency of the solving process. Additionally, the "clone" strategy allows a new worker, after requesting and dividing a partition, to duplicate the original solver and tackle the next problem. This helps to reuse learned clauses and internal solver information to accelerate problem-solving.

Our Interactive Divide-and-Conquer Environment is primarily divided into two parts. First, we encapsulate the necessary components in the `PainlessSolver` within the Painless C++ environment, derived from painless-v2,The basic configuration is based on P-CLONE-FLIP (Le Frioux et al., 2019)., and use Pybind11 to create a shared library. Then, we create a custom `PainlessEnv` in Python to facilitate interaction with the agent. `PainlessSolver` handles the solving process,

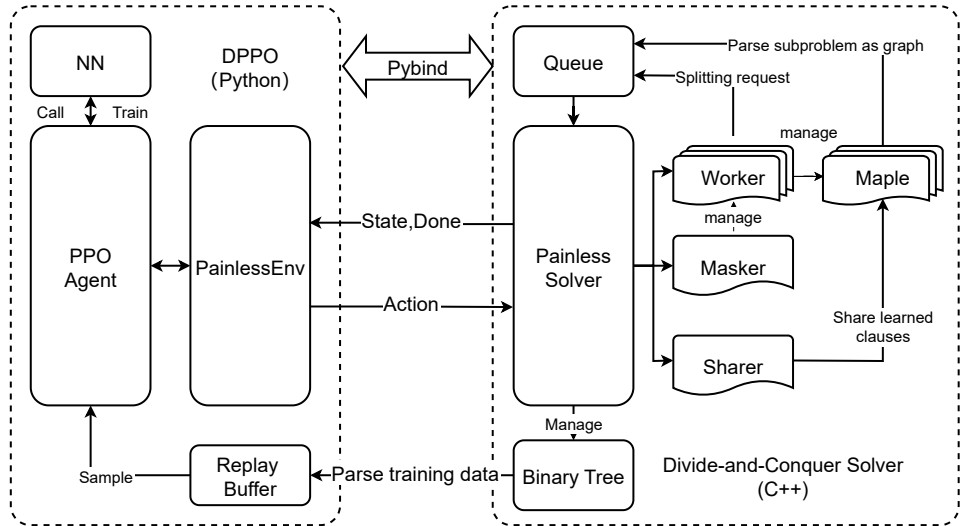

Figure 6: Interactive Divide-and-Conquer environment

while `PainlessEnv` provides the interface for the agent to interact, allowing for dynamic state transitions and reward calculations.

## B.2 PAINLESSSOLVER IN C++

---

**Algorithm 1:** PainlessSolver::step

---

**Input:** action
**Output:** state,done

1 **if** *there is a pending splitting request* **then**
2     pass action as split variable to worker;

3 **while** *not done* **do**
4     **if** *no globalEnding, no pending splitting requests, and no finished workers* **then**
5         wait for signal;
6     parse finished workers and insert as leaf nodes in binary tree;
7     **if** *there is a pending splitting request* **then**
8         state = create graph from subproblem;
9         parse worker and insert as internal node in binary tree;
10         return state, done = false;
11     **if** *globalEnding* **then**
12         return root, done = true;

---

We designed a new class, `PainlessSolver`, to handle all interactions and maintain statistics with the Python environment. When solving a new SAT problem, a `PainlessSolver` instance is created, initializing the Master thread, Worker threads, and Sharer thread, and setting the initial solving state. The Master thread manages the Worker threads, which use a CDCL solver to solve SAT problems. The Sharer thread handles the sharing of learned clauses among the Worker threads. Additionally, the `PainlessSolver` thread executes the logic described in the corresponding pseudocode. This class uses the `solve()` method to start the entire solving process. The interface built with Pybind11 enables the direct transmission of data generated by Painless to Python algorithms for processing and the reception of decision outputs from the Python environment to control the solving process. The core function `step()` of `PainlessSolver` is shown in Algorithm 1.

Due to Python's Global Interpreter Lock (GIL), only one thread can execute Python bytecode at a time. In a D&C solver, multiple workers may request splits simultaneously. To manage these requests,

we use a queue. When the master thread detects idle threads, it interrupts a worker, adds its instance to the pending split queue, and signals the `PainlessSolver` thread(line 5). The worker then blocks, waiting for the split variable. The `PainlessSolver` thread dequeues the worker instance, parses the problem state,adds it as a internal node to the binary tree,and sends the state to the Python environment(line 7-10). After processing, the `PainlessSolver` thread returns the split variable, allowing the worker to continue solving(line 2). While one worker waits, others continue solving their subproblems. Additionally, `PainlessSolver` parses finished workers and inserts them as leaf nodes in the tree(line 6). Once solving is complete, it returns the root node and sets done to true. Locks and signals ensure thread-safe communication and protect data structures.

Finally, we used Pybind11 to package `PainlessSolver` into a shared library. This allows the shared library to be directly imported into Python, enabling interaction with `PainlessSolver` through its provided interfaces.

### B.3 PAINLESSENV IN PYTHON

To facilitate interaction between the agent and the environment, we defined a custom environment similar to `gym.Env`. OpenAI's gym library provides a widely-used framework for custom reinforcement learning environments. However, `gym.Env` is designed for linear state transitions, where each state follows a sequential order, and rewards are computed based on this continuous sequence. In these environments, the `step` function returns the "next state," which is not suitable for our case. In a divide-and-conquer reinforcement learning environment, each split creates two subproblems, forming a binary tree structure rather than a direct "forward" relationship between states. Additionally, it is not possible to immediately obtain the reward in divide-and-conquer, as this information is embedded within the tree structure. To address these challenges, we created a custom class `PainlessEnv`, mimicking the basic structure of the gym library but without inheriting it. This allows us to define state transitions and reward logic flexibly, accurately handling parallel and tree-structured transitions.

`PainlessEnv` accepts a set of SAT problem paths and related configuration parameters as initial inputs, such as solver timeout, CPU core count, and levels of learned clauses. It provides `reset` and `step` methods for interaction. The `reset` method fully reinitializes the solving environment, creating a new `PainlessSolver` instance, loading a new SAT problem, and starting the solving process. The agent returns a split variable, which the `step` method passes to the `PainlessSolver` environment, then waits for the next split state while recording necessary training data. Upon solving the problem or timeout, `PainlessEnv` retrieves the root node of the binary tree, analyzes the solving process, and calculates the reward for each action. It performs a post-order traversal from the root node to compute the discounted rewards for all internal node states. This tree structure details the solving paths from root to leaf nodes, storing all this data in a replay buffer for subsequent machine learning model training.

## C DETAILS OF REINFORCED DIVIDE-AND-CONQUER

### C.1 NEURAL NETWORK

Our approach utilizes a modified Graph Neural Network (GNN) structure similar to Graph-Q-Sat, replacing the DQN structure with an Actor-Critic network. As shown in Figure 7, the neural network encodes the SAT problem information as a graph. The Encoder transforms initial features into higher-dimensional representations, the Processor performs iterative message passing between nodes, edges, and global attributes, and the Decoder outputs variable, clause, and global features. The Actor network evaluates the variable features, outputs the probabilities for selecting each variable after removing invalid split variables, and the Critic network uses clause and global features to assess the current state's value. We implemented our neural network using PyTorch(Paszke et al., 2019) and PyTorch Geometric(PyG)(Fey & Lenssen, 2019).

We adopted Graph Networks (Battaglia et al., 2018) as the basis of our GNN architecture. This framework is known for its versatility in handling various graph structures. The GN framework consists of three main parts: Encoder, Processor, and Decoder. The Encoder independently converts node, edge, and global features into higher-dimensional representations using MLPs. The Processor, the core component, updates these features through multiple layers of message passing, aggregating information from connected nodes and edges, and applying batch normalization to ensure scale

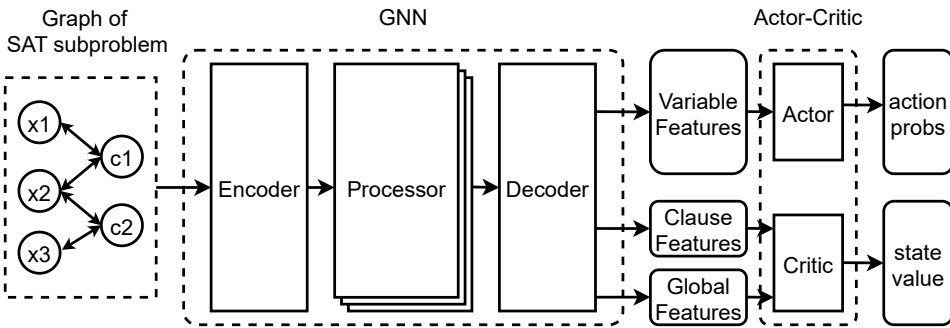

Figure 7: Network of RDC-SAT.

invariance across nodes. The Decoder simplifies the complex graph structure into actionable outputs, retaining node and global features for the subsequent PPO usage while discarding edge features. Layer normalization (Ba et al., 2016) is applied within MLPs to expedite training convergence and reduce sensitivity to input scale variations.

The Actor-Critic network leverages features derived from the GNN to perform variable selection and value estimation. The Actor network, responsible for decision-making, processes the variable features through an MLP and applies a softmax function to produce a probability distribution over the variables, excluding already chosen or invalid ones via a masking strategy. The Critic network evaluates the cumulative rewards from the current state using clause and global features. It employs attention pooling networks to aggregate clause nodes and global features, which are then fed into an MLP to generate the state value. This architecture allows the system to separate decision-making from value assessment, facilitating more precise policy adjustments and long-term reward optimization.

### C.2 SETTINGS AND HYPERPARAMETERS

We implemented a Distributed Proximal Policy Optimization (DPPO) framework, as shown in Figure 8, to accelerate data collection and training using one training process and multiple exploration processes. This framework includes a message queue for transferring collected data from exploration processes to the training process and a parameter server for synchronizing updated network parameters from the training process to the exploration processes. The message queue allows real-time transmission of states, actions, rewards, and subsequent states to the training process, which then stores them in a replay buffer for training. After updating the network parameters, the training process synchronizes these updates to the parameter server. Exploration processes check the parameter server for updates before solving new problems and update their parameters accordingly.

Table 2 shows hyperparameters were used in our experiments:

## D DATASETS

### D.1 TRAINING DATA

For our task involving reinforcement learning and graph neural networks, the most critical factors are problem size, difficulty, and structural richness. The ideal dataset should have a moderate problem size, which demands less memory during training; moderate difficulty, allowing for the rapid collection of substantial training data; and rich structure, enabling the GNN to learn more information. Although datasets like SAT Competition possess rich structural features, their generally high difficulty and larger size significantly slow down the training process for reinforcement learning. In contrast, random datasets offer highly controllable size and difficulty, and the learning clauses generated during the process can introduce additional structural features. Therefore, we chose to use a random dataset for our experiments.

Our training dataset comes from the dataset provided by the study(Cameron et al., 2020) . This study provides 11 datasets ranging from 100 to 600 variables, each containing 5000 satisfiable (SAT) and

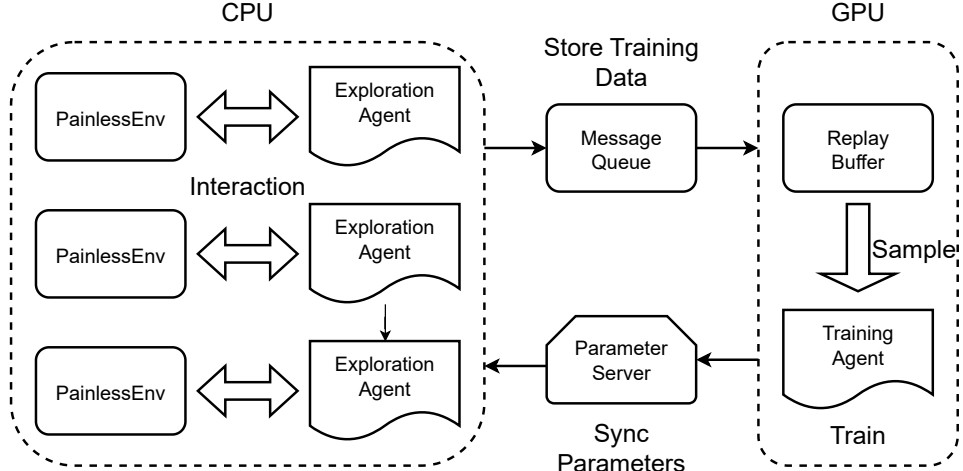

Figure 8: Architecture of. Exploration agents interact with the PainlessEnv on CPU, collect training data, and send it to the message queue. The training agent on GPU samples data from the replay buffer to update the model parameters, which are then synchronized back to the exploration agents.

5000 unsatisfiable (UNSAT) problems. We randomly selected 500 SAT and 500 UNSAT problems from the datasets with 300 and 350 variables as our training dataset. These problems have solving times ranging from a few seconds to several tens of seconds, and during the solving process, they generate thousands to tens of thousands of core and tier2 learned clauses. Consequently, the resulting graphs have relatively rich structural features.

### D.2 EVALUATION DATA

To investigate whether RDC-SAT has effectively learned the splitting heuristics, we randomly selected 200 SAT instances from the random dataset at the phase transition. This set includes 100 problems from the 450-1917 dataset and 100 from the 500-2129 dataset, with a 1:1 ratio of satisfiable to unsatisfiable instances. This dataset provides a suitable level of difficulty for the baseline solvers, enabling us to distinguish their performance.

To evaluate whether RDC-SAT, trained on random datasets, can generalize to other types of more extensive, application-oriented datasets, we focused on small to medium-sized SAT problems from the SAT Competitions dataset. Some problems in the competition are extremely large, with many SAT instances having tens of millions of clauses. Applying GNNs to these problems would consume significant memory and computational resources. For such large-scale problems, our RDC-SAT solver automatically switches to the default splitting heuristics. Therefore, we primarily focus on the performance of various methods on small to medium-sized SAT problems. We first screened 400 problems from the SAT Competition 2023 (Balyo et al., 2023) dataset and selected those with a total number of variables and clauses less than one million, resulting in 305 SAT problems. These small to medium-sized problems constitute the majority of the SAT Competition.

## E EXPERIMENT DETAILS

### E.1 BASELINES SELECTION

The Painless framework provides three splitting heuristics: based on Flips, PR, and using CDCL solver branch heuristics to select split variables. We used these three methods for our evaluations. However, in a small-scale test, we found that the PR method did not perform well on the random dataset, leading us to exclude it from our random dataset tests. Additionally, we chose not to use MaplePainlessDC as a baseline . MaplePainlessDC(Nejati et al., 2020) is also a D&C solver based on the Painless framework, which uses a machine learning method to optimize the splitting heuristic. While the performance reported in the original paper exceeds that of Treengeling, it has been observed

Table 2: Hyperparameters used in our experiments

| Parameter | Value | Description |
|---|---|---|
| **PainlessEnv Configuration** | | |
| timeout | 1000 | Maximum time (in seconds) allowed for exploration agent |
| ncpus | 16 | The number of threads used by each exploration agent |
| use_learnt | 2 | In addition to the original clause, two levels of learning clauses, core and tire2, are used to build the graph |
| node_feats | 4 | Number of node features |
| edge_feats | 2 | Number of edge features |
| global_feats | 4 | Number of global features |
| use_LRB | true | Use LRB as branch heuristic |
| **Network Configuration** | | |
| encoder_out_dims | (32, 32, 32) | Dimensions for the encoder outputs (node, edge, global) |
| core_out_dims | (64, 64, 32) | Dimensions for the core outputs (node, edge, global) |
| decoder_out_dims | (32, 1, 32) | Dimensions for the decoder outputs (node, edge, global) |
| core_steps | 4 | Number of processing steps in the Processor |
| n_hidden | 1 | Number of hidden layers in Processor's MLP |
| hidden_size | 64 | Size of hidden layers in GNN's MLP |
| e2v_aggregator | sum + BN | Aggregation method for edge to vertex. After sum aggregation, we use batch normalization(Ioffe & Szegedy, 2015) to normalize the features |
| independent_block_layers | 0 | Number of hidden layer of Encoder and Decoder |
| n_hidden_actor | 2 | Number of hidden layer of actor's MLP |
| n_hidden_critic | 2 | Number of hidden layer of critic's MLP |
| activation$_{function}$ | LeakyRelu(Maas et al., 2013) | Activation function unsed in MLP |
| **DPPO Configuration** | | |
| optimizer | Adam(Kingma & Ba, 2014) | Optimizer used |
| lr_gnn | 1e-3 | Learning rate for the GNN |
| lr_actor | 1e-3 | Learning rate for the actor network |
| lr_critic | 1e-3 | Learning rate for the critic network |
| policy_loss_weight | 1.0 | Weight of the policy loss |
| value_loss_weight | 0.5 | Weight of the value loss |
| entropy_loss_weight | 0.1 | Weight of the entropy loss |
| eps_clip | 0.2 | Clipping parameter for PPO |
| epsilon_start | 1.0 | Starting value for epsilon in epsilon-greedy exploration |
| epsilon_final | 0.01 | Final value for epsilon in epsilon-greedy exploration |
| epsilon_decay | 200 | Decay rate for epsilon |
| normalize_reward | True | Whether to normalize rewards |
| gamma | 1.0 | Discount factor for future rewards |
| train_epochs | 1 | Number of training epochs |
| train_max_steps | 10000 | Maximum number of training steps |
| sample_max_nodes | 300000 | Maximum number of nodes of one batch to sample |
| num_of_process | 5 | Number of exploration processes |

from the experimental results of SAT Competition 2020(Nejati & Ganesh) that MaplePainlessDC can produce incorrect results in a small number of SAT instances. This potential unreliability in solving performance led us to exclude MaplePainlessDC as a baseline in our evaluations.

### E.2 EXPLANATION FOR USING LRB OVER VSIDS IN RANDOM DATASET EXPERIMENTS

The decision to use different branching heuristics for random and SAT Competition benchmarks stems from the observed poor performance of VSIDS on random datasets, as commonly noted in empirical studies. We conducted a small experiment using VSIDS as the branching heuristic on 100 instances from the 400-1704 random dataset with a time limit of 1000 seconds. The results of this experiment are presented in Table 3. This performance is significantly worse compared to using LRB as the branching heuristic (see Table 4 for a comparison). Therefore, we opted for Maple's LRB branching heuristics for both training and evaluation on the random datasets. It is also worth noting that even with VSIDS, RDC-SAT still significantly outperforms the baseline.

Table 3: Performance Comparison of Splitting Heuristics on 400-1704 random dataset with VSIDS as Branching Heuristics

| Splitting Heuristics | Clock time (s) | PAR2 | Solved |
|---|---|---|---|
| RDC-GPU-always | **388.23** | **588.18** | **80** |
| RDC-CPU-once | 405.48 | 703.34 | 77 |
| FLIPS | 460.70 | 720.72 | 74 |
| VSIDS | 502.28 | 852.20 | 65 |

### E.3   MORE EVALUATION RESULT ON RANDOM DATASET

Table 4 shows a detailed comparison of various splitting heuristics across different datasets. This table includes all instances, including those that no solver could solve within the time limit.

Firstly, for relatively easy SAT problems, such as those in the 350-1491 dataset, using the default splitting heuristics like FLIPS and LRB can yield quick results. In these cases, the overhead of using a GNN-based approach might actually slow down the process. For example, FLIPS and LRB both performed better than RDC-GPU-always and RDC-CPU-once in terms of average solving time. This is because the simpler problems do not require the sophisticated analysis that GNN provides, and the additional computation introduces unnecessary delay. However, as the problem difficulty increases, the advantages of RDC-SAT become more apparent. In the 450-1917 dataset, RDC-GPU-always significantly outperformed traditional methods, with a total average solving time of 526.21 seconds compared to FLIPS' 827.07 seconds and LRB's 780.34 seconds. on the most challenging dataset, 500-2129, RDC-GPU-always not only achieved a lower average solving time but also solved more instances than the baselines. Specifically, RDC-GPU-always solved 30 instances in this dataset, whereas the best baseline method solved only 26 instances, representing a 15.38% increase in the number of solved instances.

Additionally, the table reveals that the PR splitting heuristic performed very poorly on the 350-1491 dataset, with an average solving time of 61.65 seconds. This performance suggests that the PR method may not be suitable for solving random SAT problems. Due to this, we did not test the PR method on larger datasets.

Table 4: Comparison of splitting Heuristics across datasets

| Dataset | Splitting Heuristics | SAT Avg (s) | #SAT | UNSAT Avg (s) | #UNSAT | Total Avg (s) | PAR2 |
|---|---|---|---|---|---|---|---|
| random 350-1491 | RDC-GPU-always | 12.34 | 50 | 19.80 | 50 | 16.03 | 16.03 |
| | RDC-CPU-once | 11.16 | 50 | 17.99 | 50 | 14.58 | 14.58 |
| | FLIPS | **10.76** | 50 | **17.22** | 50 | **13.99** | **13.99** |
| | LRB | 11.02 | 50 | 17.96 | 50 | 14.49 | 14.49 |
| | PR | 22.36 | 50 | 103.83 | 50 | 61.65 | 61.65 |
| random 400-1704 | RDC-GPU-always | 19.14 | 50 | **63.38** | 50 | **41.27** | **41.27** |
| | RDC-CPU-once | 18.29 | 50 | 64.35 | 50 | 41.32 | 41.32 |
| | FLIPS | 18.16 | 50 | 80.27 | 50 | 49.21 | 49.21 |
| | LRB | **17.90** | 50 | 67.84 | 50 | 42.87 | 42.87 |
| random 450-1917 | RDC-GPU-always | **291.48** | 50 | **760.94** | **49** | **526.21** | **576.21** |
| | RDC-CPU-once | 353.63 | 50 | 835.91 | **49** | 594.75 | 644.77 |
| | FLIPS | 390.83 | 50 | 1263.31 | 48 | 827.07 | 927.07 |
| | LRB | 341.64 | 50 | 1219.04 | 47 | 780.34 | 930.34 |
| random 500-2129 | RDC-GPU-always | **3072.05** | **26** | **4777.20** | **4** | **3924.63** | **7424.63** |
| | RDC-CPU-once | 3397.68 | 23 | 4804.05 | **4** | 4100.86 | 7750.86 |
| | FLIPS | 3255.54 | 24 | 4898.27 | 2 | 4076.91 | 7776.91 |
| | LRB | 3627.84 | 20 | 4898.62 | 3 | 4263.23 | 8113.23 |

### E.4   DOMAIN-SPECIFIC BENCHMARK COMPARISON OF RDC-SAT AND SOTA PORTFOLIO SOLVERS

SAT Competition results indicate that D&C solvers generally lag behind portfolio approaches in average performance. However, D&C solvers demonstrate unique advantages in specific scenarios.

For instance, we conducted a small-scale experiment on a dataset containing 100 randomly selected instances from the miter benchmarks, which are used to verify circuit equivalence. The timeout was set to 1000 seconds, with all methods using 16 threads and VSIDS as the branch heuristic for the Sequential Solver. Using the model trained in our work. Table 5 compares our RDC-SAT with the D&C solver FLIPS (Le Frioux et al., 2019) and PRS (Chen et al., 2023) – the SAT Competition 2023 Parallel Track champion portfolio solver.

Table 5: Performance Comparison on Circuit Equivalence Verification

| Method | Solved | Avg. Time (s) | PAR2 |
|---|---|---|---|
| RDC-SAT (D&C) | **80** | **313.64** | **513.64** |
| FLIPS (D&C) | 76 | 333.66 | 573.66 |
| PRS (Portfolio) | 79 | 385.44 | 595.44 |

This result shows that D&C solvers can outperform Portfolio solvers in specific domains. In this experiment, RDC-SAT solved the most instances and achieved the lowest PAR2, reducing PAR2 by 13.74% compared to the SOTA Portfolio solver PRS. The RDC-SAT model, trained on a random dataset, generalized effectively to the circuit equivalence dataset.

## F    THREATS TO VALIDITY

During the training phase of our D&C SAT solver using reinforcement learning, several potential threats to validity may impact the results. Firstly, due to Python's Global Interpreter Lock (GIL) enforcing single-threaded execution for Python bytecode, we use a message queue to manage all split requests. When multiple threads simultaneously request splits, only one worker can call the neural network to obtain the split point, while other threads enter a waiting state. This restriction may lead to inefficiencies and does not fully replicate the actual D&C solver's runtime environment. Additionally, during training, multiple threads simultaneously calling the neural network, along with the training process itself, can cause GPU resource contention, leading to delays in split decisions and affecting training performance. Consequently, we opted to use CPUs for neural network calls during training to avoid GPU contention. However, this results in overestimated split times during training, as it does not accurately reflect the performance of the D&C solver in a real-world environment. Furthermore, our current setup limits the efficient use of computational resources, particularly GPUs. If more computational resources were available, such as additional GPUs for running the DPPO framework, this issue could be significantly mitigated. Enhanced computational resources would allow for more realistic training conditions, better reflecting the actual operational environment of the D&C solver, and potentially leading to more effective learning outcomes. These factors may result in the model's performance not reaching its optimal potential. With more computational resources to train the model and better simulate actual environments, RDC-SAT could achieve better results.

Another potential threat to validity pertains to the experimental evaluation of our solvers. All experiments were conducted on a machine equipped with a CPU featuring 144 logical cores, an A100 GPU, and 256 GB of RAM. Due to the extensive time required for the experiments, we executed multiple solvers in parallel, allocating 16 threads to each solver. For the RDC-CPU-once configuration, the neural network calls were restricted to using a maximum of 16 threads. Although the number of threads used per solver is less than the total logical cores, the total number of threads during the SAT Competition experiments exceeded the number of physical cores, potentially leading to resource contention. This contention could affect performance, but it should impact all solvers equally, thus maintaining fairness in the comparisons. We primarily used small to medium-sized SAT problems, so we did not observe significant memory contention.

