# OpenReview forum: "Learning Splitting Heuristics in Divide-and-Conquer SAT Solvers with Reinforcement Learning"
_ICLR.cc/2025/Conference — ICLR 2025 Poster_

### Official Review · Reviewer_DJMo · 2024-10-27

**Soundness:** 3
**Presentation:** 4
**Contribution:** 2
**Rating:** 8
**Confidence:** 4

**Summary:**

This paper introduces a new SAT solver, RDC-SAT, which combines reinforcement learning with a divide-and-conquer strategy. Currently, modern SAT solvers are using parallelization strategies to enhance computational efficiency. While portfolio-based solvers lead in SOTA performance, the divide-and-conquer strategy still retains competitive potential due to its structure-guided nature. Similar to the step-by-step approach in divide-and-conquer, reinforcement learning also has the ability to make sequential decisions under a MPC framework. The authors leverage this motivation, formalizing the D&C splitting process within the Painless framework into an RL environment with a designed reward function, and train it using a DPPO policy. Two versions of RDC-SAT, using different device-utilization approaches, along with traditional splitting strategies are tested on a random dataset and the SAT competition 2023 dataset. The evaluation results show that the RDC-SAT solver has optimized computational efficiency and better generalization compared to other strategies.

**Strengths:**

1. The motivation for combining D&C SAT solver and RL approach is clear and reasonable. It is a valuable A+B attemptation.
2. This work do not simply focus on pushing forward SOTA Portfolio-based solvers; instead, it reflects on the potential of the D&C strategy with explainable points. It introduces more thoughtful topics to the research community.
3. The methodology flow presented in chapter 3-4 is well-structured. Readers can easily follow the construction of the entire work in a sequential way. Detailed definitions are also presented well.
4. The authors provide sufficient detailed information in the appendix for a comprehensive presentation, including related works, detailed algorithms, datasets, and so on. This greatly aids understanding and reproduction.

**Weaknesses:**

1. This is an A+B work, and it does not push forward SOTA solver performance.
2. There are details remain unclear in time, space & scale when using GNNs. (See questions 1-3)
3. There are details remain unclear in reinforcement learning settings. (See questions 4)
4. The experiments could be more comprehensive. (See question 5-8)
5. Other minor concerns. (See question 9-10)

**Questions:**

I didn't read all of the appendix, so if there is anything I've missed, please let me know.
1. During the evaluation stage, if the solver needs to generate a new GNN every time a splitting strategy is applied (due to the new or deleted learned clauses/variables etc), doesn't it cost a lot both in time and space compared to tranditional strategies? Can you show me the experimental statistic on this?
2. Replay buffer: Also, in RL training, data stored in graph could cost a lot of space. Is there any consideration of this?
3. The paper mentioned that during the splitting process, the learned clauses can expand the original scale from k+ to 10-50 k+ clauses. This puts more pressure on time and space concerns. How does RDC-SAT manage this? I believe these three questions may relate to the CPU-once settings.
4. The paper did not give a clear definition of the RL action, apart from the state and reward. The splitting stragegy needs a chosen variable to split; is the chosen variable denoted as an action? This should be well-defined in the main paper.
5. The detailed definition of PAR2 metric should be introduced in the experiments.
6. The experiments only compare to the trandition splitting methods. Are there any learning-based splitting methods to compare with?
7. Do the chosen tranditional splitting methods perform a SOTA level in D&C solvers? If not, what's the gap between RDC-SAT and SOTA D&C solver? Moreover, what's the gap between RDC-SAT and SOTA SAT solvers (including Portfolio)? There should be a promising future where, even though currently D&C solvers are not SOTA, they could become SOTA with a relatively small gap to overcome.
8. Can you train RDC-SAT on the SAT competition Dataset and show the in-distribution evaluation result? It may prove the direct applicability in industrial applications.
9. What does [1, 0, ...] mean in Figure 2?
10. On page 6, line 275, why do you use "absolute" values? Since the defined value here is time, the value is certainly non-negative.

---

> ### Author Response · Authors · 2024-11-20
> **Response to Reviewer DJMo (1/2)**
>
> We appreciate the feedback provided by Reviewer DJMo. We'd like to respond to your questions as follows:
>
> # Response to questions(Questions)
>
> **Q1. During the evaluation stage, if the solver needs to generate a new GNN every time a splitting strategy is applied (due to the new or deleted learned clauses/variables, etc.), doesn't it cost a lot both in time and space compared to traditional strategies? Can you show me the experimental statistic on this?**
>
> During evaluation, the GNN model is loaded only once and reused for all splits. It can handle SAT problem graphs with different variable and clause counts, so each split only requires creating a new graph representation, not rebuilding the model. As shown in Table 1 and Figure 5 in the paper, the splitting overhead in RDC-SAT is relatively small. Even in the RDC-GPU-always setting, where the GNN is invoked at every split, the CPU time spent on splitting accounts for only 0.12% and 1.33% of the total solving time on the random dataset and SAT Competition dataset, respectively. Despite this minimal overhead, the splitting heuristic significantly reduces the overall solving time.
>
>
> **Q2. Replay buffer: Also, in RL training, data stored in graph could cost a lot of space. Is there any consideration of this?**
>
> Yes, storing graph data in the replay buffer does consume space. After these graphs are used for training, we immediately delete those associated with each SAT problem. Additionally, we balance the rate at which graphs are generated during exploration and consumed during training by adjusting process parameters to ensure efficient memory usage. This approach has effectively kept memory usage under control.
>
> **Q3. The paper mentioned that during the splitting process, the learned clauses can expand the original scale from k+ to 10-50 k+ clauses. This puts more pressure on time and space concerns. How does RDC-SAT manage this? I believe these three questions may relate to the CPU-once settings.**
>
> The expansion of learned clauses is related to clause learning in the CDCL process, not the D&C algorithm. This phenomenon is particularly pronounced in SAT problems near the phase transition due to the high complexity of the solution space. At this threshold, problems exhibit frequent conflicts as the boundary between satisfiability and unsatisfiability is extremely narrow. These frequent conflicts trigger more clause learning in the CDCL process, leading to a rapid increase in the number of learned clauses. This also enables the model to learn richer structural features.
>
>
> **Q4. The paper did not give a clear definition of the RL action, apart from the state and reward. The splitting strategy needs a chosen variable to split; is the chosen variable denoted as an action? This should be well-defined in the main paper.**
>
> Yes, in our RL setup, the action is defined as the selection of a variable to split, excluding variables that have already been eliminated or previously chosen in earlier splits.
>
> **Q5. The detailed definition of the PAR2 metric should be introduced in the experiments.**
>
> PAR2 is a widely used metric in SAT solving, assigning twice the timeout limit to unsolved instances. We will include a detailed explanation of PAR2 in future revisions to improve clarity.

---

> > ### Comment · Reviewer_DJMo · 2024-11-21
> >
> > Thank you for the explanation. I have some further comments below:
> > 1. Sorry for the wrong statement "GNN" in **Q1**, my true meaning is "graph". I have understand that the proper splitting frequency could reduce the burden for the model to generate graph.
> > 2. According to the response of **Q2**, you said "After these graphs are used for training, we immediately delete those associated with each SAT problem.".Does this mean the data-collecting process (D&C solver) and the RL training process are performing simultneously?
> > 3. I would be appreciate if the definition of the RL action in **Q4** is well-defined in the main paper. Currently I do not see a clear definition in the paper.
> > 4. Following the discussion in **Q7**, what does PRS perform on the main experiment setting in this paper?
> > 5. I am looking forward to the further evaluation on EDA and cryptanalysis scenarios mentioned in **Q8**. However, I still recommend to evaluate RDC-SAT on the SAT Competition Dataset, its subset or other real world datasets, rather than the random 3-SAT dataset, for a stronger evaulation support. Also, I believe the circuit equivalence dataset result (in your response to Reviewer 5BwV) can be completed and added into the paper.
> > 6. I have understand your explanation in response to **Q9**. But the clause nodes in Figure 2 still got 1 in the first position, rather than the second position. Is this a typo? Also, what about the [0, 1] beside each edge? Moreover, I suggest you add the explanation into the caption of Figure 2 to improve clarity.

---

> > > ### Author Response · Authors · 2024-11-25
> > >
> > > We appreciate the reviewer’s follow-up questions and valuable insights. In our Global Official Comment, we clarified the primary goals and constraints of our experimental design. Here, we provide further detailed responses to specific points raised.
> > >
> > > # Response to Further Comments
> > >
> > > **Q2. Does this mean the data-collecting process (D&C solver) and the RL training process are performed simultaneously?**
> > >
> > > Yes, during the training process with DPPO, we simultaneously run multiple exploration processes and one training process. The exploration processes solve SAT problems and synchronize the collected training data (binary trees containing graph information, solving times, satisfiability status, etc.) with the training process. The training process extracts the required information from these trees for training, updates the RL model, and deletes the processed data. Once updated, the model parameters are synchronized with the exploration processes. This process is performed concurrently.
> > >
> > > **Q7. What does PRS perform on the main experiment setting in this paper?**
> > >
> > > Due to the time constraints during the discussion period, we were unable to conduct a full evaluation of PRS under our experimental setup. Additionally, we have clarified our rationale for baseline selection in the Global Official Comment. In future work, we plan to conduct a more extensive evaluation, including comparisons with Portfolio and other solvers, over a broader range of datasets and setups.
> > >
> > > **Q8. Evaluating RDC-SAT on SAT Competition datasets or subsets instead of random 3-SAT datasets**
> > >
> > > We fully agree that evaluating RDC-SAT on subsets of the SAT Competition dataset or other real-world datasets would provide stronger evidence of its effectiveness, and this will be a key focus of our future work. In our current paper, the experiments are primarily designed to demonstrate the effectiveness and generalization capability of RDC-SAT. Therefore, we only used random datasets for training. Even so, the experimental results show that RDC-SAT, when trained on small random datasets, can generalize well to application datasets such as those from the SAT Competition.
> > >
> > > **Q9. Clarifications on Figure 2**
> > >
> > > Thank you for pointing out this issue. The clause nodes in Figure 2 incorrectly have "1" in the first position instead of the second position, which is a typo. The `[0, 1]` next to each edge represents the initial edge features in the graph, indicating the polarity of the variable in the clause. Specifically, `[0, 1]` denotes a positive literal (e.g., `x`), while `[1, 0]` denotes a negative literal (e.g., `¬x`). We have corrected this error and added further explanations in the revised manuscript.
> > >
> > > # Other minor Suggestions
> > >
> > > We sincerely appreciate your constructive suggestions regarding the presentation and experiments in our paper. In the revised version, we have already addressed aspects such as the definition of actions and graph features to improve clarity and detail.

---

> > > > ### Comment · Reviewer_DJMo · 2024-11-26
> > > >
> > > > Thank you for your thorough responses to the additional queries. I appreciate the effort and find your explanations satisfactory. I am content to keep the current score and eagerly anticipate future developments from this work.

---

> > > > > ### Author Response · Authors · 2024-12-02
> > > > >
> > > > > Thank you for your kind and encouraging feedback. We are delighted to hear that our explanations were satisfactory and that our work met your expectations. We look forward to further advancing this work and making greater contributions to the field. Once again, we deeply appreciate the time and effort you have dedicated to thoroughly reviewing our paper and providing such valuable feedback and suggestions.

---

> ### Author Response · Authors · 2024-11-20
> **Response to Reviewer DJMo (2/2)**
>
> **Q6. The experiments only compare to the traditional splitting methods. Are there any learning-based splitting methods to compare with?**
>
> We have explained the selection of splitting method baselines in Appendix E.1 (lines 1020–1067), there are no suitable learning-based splitting methods available to use as baselines for our experiments.
>
> **Q7. Do the chosen traditional splitting methods perform at a SOTA level in D&C solvers? If not, what's the gap between RDC-SAT and SOTA D&C solvers? Moreover, what's the gap between RDC-SAT and SOTA SAT solvers (including Portfolio)? There should be a promising future where, even though currently D&C solvers are not SOTA, they could become SOTA with a relatively small gap to overcome.**
>
> Based on results in the paper[1], the Painless D&C solver using FLIPS as the splitting heuristic outperformed both MapleAmpharos and Treengeling on the SAT Competition dataset, indicating that it can be considered a SOTA D&C SAT solver. From the results of recent SAT Competitions, Portfolio solvers generally outperform D&C solvers in average performance. However, D&C solvers exhibit unique advantages in specific domains. For example, in our experiment on the circuit equivalence dataset, RDC-SAT solved the most instances and achieved a PAR2 that is 13.74% lower than that of the SOTA Portfolio solver PRS (as detailed in our response to Reviewer 5BwV). In addition to improving the Splitting Heuristic, we are actively working on further enhancements to RDC-SAT to narrow the gap and bring its performance closer to that of SOTA Portfolio solvers.
>
> **Q8. Can you train RDC-SAT on the SAT Competition Dataset and show the in-distribution evaluation result? It may prove the direct applicability in industrial applications.**
>
> Training directly on the SAT Competition dataset requires significantly more time due to the higher average complexity of the problems. However, we believe that training on this dataset could enhance RDC-SAT’s performance on industrial application datasets. We are currently collecting datasets in domains such as EDA and cryptanalysis and plan to train the model on these datasets and evaluate its performance in these practical application domains in future work.
>
> **Q9. What does [1, 0, ...] mean in Figure 2?**
>
> The [1, 0, ...] represents the initial feature values assigned to each node in the graph. The first position being `1` indicates that the node is a variable node, while the second position being `1` would indicate that the node is a clause node. The subsequent positions represent normalized features of variables or clauses, such as variable activity scores or clause LBD (Literal Block Distance).
>
> **Q10. On page 6, line 275, why do you use "absolute" values? Since the defined value here is time, the value is certainly non-negative.**
>
> The "absolute" value here refers to the discounted reward \( R \). In our setup, the discounted reward can be seen as an estimate of the negative solving time, so maximizing the reward is equivalent to minimizing the solving time.
>
> References:
>
> [1]Le Frioux et al. Modular and efficient divide-and-conquer SAT solver on top of the painless framework.

---

### Official Review · Reviewer_uKwC · 2024-11-01

**Soundness:** 3
**Presentation:** 3
**Contribution:** 3
**Rating:** 6
**Confidence:** 4

**Summary:**

The paper presents a framework that employs deep reinforcement learning to enhance splitting heuristics in divide-and-conquer SAT solvers. By utilizing a graph representation to dynamically extract features from the solving state, RDC-SAT optimally selects split variables through a reinforcement learning paradigm. The method addresses the challenges posed by tree-like state transitions in divide-and-conquer scenarios and introduces tailored reward functions for both satisfiable and unsatisfiable SAT problems. In the experiment section, the proposed method is trained on random 3CNF at threshold instances and helps CDCL solvers achieve better performance. The trained model generalizes to industrial instances as well.

**Strengths:**

1. Improving the performance of parallel SAT solvers, especially D&C solvers, is one of the important tasks in SAT solving.  D and C solvers are the paper did an integration of reinforcement learning and SAT solving. The combination of RL and D&C solvers is a solid research direction.

2. It is very interesting to see that models trained on random instances can be generalized to instances from SAT Competition. I believe that it implies that there exists certain rules/techniques in SAT solving that are sharable between different types of instances.
It is not clear whether things learned from random instances can be generalized into industrial instances.
the size of problems are relatively small, does it generalize to larger instances?

3. This paper is well written with methodologies clearly explained.

**Weaknesses:**

1. Exactly why the model trained on just random 3CNF can generalize on industrial benchmarks is still mysterious to me. I believe this point is important and worth more investigation. Can the heuristics learned by the model be extracted and represented symbolically? Does learning on industrial benchmarks further improve the performance on instances from SAT competition?

2. The size of random instances used in training and testing is relatively small (300-500), though they are indeed on the threshold of satisfiability. I am not sure whether those problems are actually considered challenging to CDCL solvers. It would be helpful to see if the approach generalizes to larger random instances with thousands of variables.

**Questions:**

See the above

---

> ### Author Response · Authors · 2024-11-20
>
> We appreciate the feedback provided by Reviewer uKwC. We'd like to respond to your questions as follows:
> # Generalization from Random 3CNF to Industrial Benchmarks(Weakness 1)
> Although the initial structural features of random 3CNF problems differ significantly from industrial benchmarks, the clause learning process in CDCL solvers generates many additional clauses during solving. These learned clauses introduce diverse structural features that the model can leverage to identify universal patterns in SAT problems. Additionally, beyond structural features, our model incorporates initial features such as variable activity and clause LBD (Literal Block Distance), which further help the model capture generalizable patterns. This combination of learned and initial features likely explains why the model trained on random 3CNF problems generalizes effectively to industrial benchmarks. Furthermore, we are conducting additional experiments in domains such as EDA (Electronic Design Automation) and cryptanalysis to evaluate the model’s performance in diverse industrial applications. We anticipate that training on industrial benchmarks will further enhance the model’s capabilities, and we plan to explore this direction in future work.
>
> Due to the non-linear nature of the GNN-based framework, extracting and symbolically representing the learned heuristics is challenging. While direct symbolic representation may not be feasible, exploring methods to interpret and analyze the behavior of the learned heuristics in specific scenarios remains a valuable research direction.
>
> # Problem Size and Generalization to Larger Instances(Weakness 2)
> The reviewer raised concerns about the relatively small size of the random instances used in training and testing (300-500 variables). However, the size of a SAT problem does not necessarily correlate with its solving difficulty. For example, in SAT Competition 2023, no solver was able to solve a 329-variable cliquecolouring problem (hash: 4408f30395fb799e01b34b0dc7ee1d62) within the 5000-second timeout. In contrast, many larger problems with hundreds of thousands of variables can be solved within seconds.
>
> The random instances we used were specifically selected at the phase transition threshold, which is known to be the most challenging region for 3-SAT problems. This threshold represents a critical clause-to-variable ratio, where the probability of satisfiability changes abruptly, leading to a peak in computational difficulty. In our experiments, CDCL solvers frequently timed out on phase transition instances with 450 variables and 1917 clauses, particularly on unsatisfiable cases. This demonstrates that the dataset we used is highly challenging for SAT solvers. For less challenging, non-phase transition problems, RDC-SAT is expected to generalize well to larger instances. This is supported by its performance on the SAT Competition dataset, which includes problems with hundreds of thousands of variables and clauses.

---

> > ### Comment · Reviewer_uKwC · 2024-11-21
> >
> > Thanks for the response.
> >
> > For the first point, it is nice to see the intuition but it would be even better to validate those ideas with formalized concepts and experiments (maybe a future direction).
> >
> > For the second point, I agree that there can be small SAT instances that can be hard for all solvers. However, that does not mean random 3CNF, a specific class of SAT instances, can produce hard instances with small problem sizes. The threshold 4.26 is only of asymptotic significance, which I believe hardly applies to problems with only 300-500 variables. Even if CDCL solvers find it hard to solve those problems, a state-of-the-art local search solver such as probsat can solve them easily. As an incomplete approach, the proposed solver should be compared to local search solvers to be fair.

---

> > > ### Author Response · Authors · 2024-11-25
> > >
> > > Thank you for your response. We will address your concerns by clarifying the comparison with local search solvers and evaluating the difficulty of our random experimental datasets through a small experiment.
> > >
> > > # Comparison with Local Search Solvers
> > >
> > > To clarify, RDC-SAT is a complete solver, capable of solving both satisfiable (SAT) and unsatisfiable (UNSAT) problems while providing formal proofs for UNSAT cases. In contrast, local search solvers like probsat are incomplete solvers, which excel at solving satisfiable random SAT problems but cannot determine unsatisfiability or provide proofs. For UNSAT problems, such solvers return `UNKNOWN`. While probsat is highly effective for satisfiable cases, its inability to address UNSAT problems makes it unsuitable for direct comparison with complete solvers like RDC-SAT.
> > >
> > > # Difficulty of random Datasets
> > >
> > > The random datasets used in our experiments are derived from [1], which states that instances with 300 variables take several seconds to solve, while instances with 600 variables can take several hours. The unsatisfiable instances within this dataset are even more challenging. To evaluate the difficulty of these instances, we conducted a small experiment using kissat, under its default configuration. We evaluated two datasets: one consisting of 50 instances with 400 variables and 1704 clauses, and another consisting of 50 instances with 450 variables and 1917 clauses. Both datasets contain only unsatisfiable instances, with a timeout of 5000 seconds for each instance. For the 400-1704 dataset, kissat solved only 3 out of 50 instances, with an average solving time of 3813.3 seconds. For the 450-1917 dataset, kissat failed to solve any instance within the timeout. Furthermore, local search solvers like probsat, being incomplete solvers, cannot solve unsatisfiable problems and would return `UNKNOWN` for all these instances These results highlight the challenging nature of our datasets.
> > >
> > > We hope this explanation clarifies your concerns and provides further insight into our work.
> > >
> > > References:
> > >
> > > [1] Cameron et al. Predicting propositional satisfiability via end-to-end learning.

---

### Official Review · Reviewer_QJyP · 2024-11-02

**Soundness:** 3
**Presentation:** 2
**Contribution:** 2
**Rating:** 3
**Confidence:** 5

**Summary:**

This paper proposes using deep reinforcement learning to enhance divide-and-conquer (D&C) based parallel SAT solving. The basic idea is to train a Graph Neural Network (GNN) model using reinforcement learning to predict splitting heuristics. The proposed method shows improved performance over common heuristics like FLIPS, VSIDS, and PR on random 3-SAT problems and SAT Competition 2023 benchmarks.

**Strengths:**

The paper addresses an important and interesting problem. The proposed approach is conceptually sound, and the results appear promising.

**Weaknesses:**

The novelty of this paper is not clearly explained. It would be beneficial to clearly highlight the incremental contribution of this work, given the existing research on RL and GNN applications in SAT solving. A thorough comparison with Graph-Q-SAT and other related work, could help emphasize the unique aspects of this approach.

While the paper aims to improve D&C-based parallel SAT solving, the methodology does not seem to include specific elements which has a clear connection with parallel SAT solving. The proposed optimization and improvement seems purely in sequential manner. It would be helpful to clarify how the proposed technique applies to parallel SAT solving.

The evaluation appears to be incomplete. The evaluation could be strengthened by including comparisons with existing parallel and sequential SAT solvers. Considering a broader range of problem sizes and incorporating benchmarks from the latest SAT competition 2024 would provide a more comprehensive assessment.

Minor Suggestions:

Section 3.1 could focus more on methodology rather than implementation details of the Gym environment.

Diversifying the training data beyond 3-SAT problems might improve the model's prediction accuracy across various SAT problem types.

**Questions:**

What is the incremental contribution of this paper?

---

> ### Author Response · Authors · 2024-11-20
>
> We appreciate the feedback provided by Reviewer QJyP. However, there appear to be several key misunderstandings regarding the content and contributions of our paper. Below, we address these concerns point by point.
>
> # Comparison with Graph-Q-SAT(Weakness 1)
> The reviewer expressed concerns about the novelty of our work and requested a comparison with Graph-Q-SAT. To clarify, RDC-SAT and Graph-Q-SAT address fundamentally different problems and applications:
>
> - **Graph-Q-SAT**: This method optimizes branch heuristics (e.g., VSIDS) for CDCL sequential solvers and is implemented on top of Minisat. It focuses on improving decision-making in a sequential solving context.
> - **RDC-SAT**: Our work optimizes the splitting heuristic in Divide-and-Conquer (D&C) SAT solvers, implemented within the Painless D&C framework. Splitting heuristics determine how problems are divided into subproblems to enable parallel solving.
>
> Although both approaches employ a GNN structure, their goals, methods, and application contexts are distinct. Graph-Q-SAT is tailored to sequential solvers, while RDC-SAT explicitly targets parallel solving efficiency within a D&C framework. As a result, a direct comparison with Graph-Q-SAT is not neccesary.
>
> # Connection to Parallel SAT Solving(Weakness 2)
>
> The reviewer suggested that our methodology lacked a clear connection to parallel SAT solving and appeared purely sequential. This is a misunderstanding of the D&C approach. The D&C framework is inherently parallel, as it recursively divides SAT problems into smaller subproblems that can be solved concurrently using multiple threads. The splitting heuristic in RDC-SAT determines how a problem is divided into subproblems, which improves parallel solving efficiency. Therefore, RDC-SAT is explicitly designed to improve parallel solving within the D&C framework.
>
> # Scope of the Evaluation(Weakness 3)
>
> The reviewer mentioned that the evaluation seemed incomplete without comparisons to parallel and sequential SAT solvers. The primary goal of RDC-SAT is to optimize the splitting heuristic in D&C solvers, so our evaluation focused on comparing RDC-SAT with splitting heuristics in the Painless D&C framework. Comparing RDC-SAT with sequential SAT solvers would not evaluate the contribution or impact of the splitting heuristic. Additionally, we are currently evaluating RDC-SAT on more datasets and plan to include additional experimental results in future versions.
>
> # Incremental Contribution of This Paper(Question)
>
> To the best of our knowledge, this is the first work to apply reinforcement learning to optimize Divide-and-Conquer (D&C) algorithms. Unlike traditional Markov Decision Processes, D&C algorithms involve tree-like state transitions, which are not directly compatible with reinforcement learning frameworks designed for linear state transitions. To optimize the splitting heuristic in D&C SAT solvers, we developed a reinforcement learning environment tailored to SAT problems, capable of handling tree-structured state transitions. We defined reward computation methods suited to D&C scenarios and designed specific reward functions based on the characteristics of SAT problems. By integrating reinforcement learning with Graph Neural Networks (GNNs), we optimized the splitting heuristic in D&C SAT solvers, enabling more efficient problem decomposition and parallel solving. Evaluations show that RDC-SAT significantly outperforms traditional splitting heuristics and effectively generalizes from random 3-SAT problems to industrial benchmarks. These results highlight the potential of our approach to improve both the performance and scalability of D&C solvers.

---

> > ### Comment · Reviewer_QJyP · 2024-11-22
> >
> > Thank you for your response. After careful consideration, I still find the level of novelty and contributions insufficient to warrant acceptance at this time. In particular, the work does not clearly demonstrate a significant improvement over SOTA solvers on the latest standard benchmarks, and the contributions of the proposed RL approach remain mediocre. Given these considerations, I will need to maintain my current score. Thank you for your understanding.

---

> > > ### Author Response · Authors · 2024-11-25
> > >
> > > Thanks for your response. The reviewer raised concerns that RDC-SAT does not match the performance of SOTA Portfolio solvers and, as a result, deemed the work lacking in novelty and its contributions to be insufficient. Below, we address these issues in more detail.
> > >
> > > # Novelty and Contributions
> > >
> > > Our work is the first to apply reinforcement learning (RL) to optimize the splitting heuristic in Divide-and-Conquer (D&C) SAT solvers and also the first to address the challenges posed by tree-like state transitions in RL. Compared to traditional heuristic methods, RDC-SAT leverages structural features to select appropriate split variables, significantly improving the performance of splitting heuristics. While our method does not elevate D&C solvers to the level of SOTA Portfolio solvers, it substantially enhances the performance of D&C solvers, achieving SOTA performance within this category. Furthermore, our method offers valuable insights for future RL applications in D&C algorithms.
> > >
> > > # Comparison with SOTA Portfolio Solvers
> > >
> > > In recent years, Portfolio solvers have outpaced D&C solvers in development. Elevating the performance of D&C solvers to higher levels requires innovative and impactful efforts. Our work advances D&C solvers by improving their splitting heuristics, enabling RDC-SAT to achieve SOTA performance within this category. To dismiss our contributions solely because RDC-SAT does not match the performance of SOTA Portfolio solvers would be unfair. D&C solvers have distinct goals and advantages, particularly in distributed and scalable environments. Although these aspects were not the primary focus of our current evaluation, they remain promising directions for future work.
> > >
> > > We hope this explanation clarifies your concerns and provides further insight into our work.

---

### Official Review · Reviewer_5BwV · 2024-11-04

**Soundness:** 3
**Presentation:** 3
**Contribution:** 2
**Rating:** 6
**Confidence:** 4

**Summary:**

This paper proposes a reinforcement learning-based (RL) splitting heuristic for Divide-and-Conquer (D&C) parallel SAT solvers. Similar ideas were applied to branching heuristics but suffered from excessive prediction calls. Benefiting from the low invocation frequency, the RL-empowered splitting heuristic can effectively reduce the overall runtime of D&C SAT solvers. The empirical evaluation over random 3-SAT and SAT competition benchmarks showed that the resulting solver, RDC-SAT outperformed the traditional D&C solvers.

**Strengths:**

1. This work leverages the low invocation frequency of the splitting heuristic for D&C SAT solvers and improves the overall runtime performance. In contrast, the previous works on branching heuristics failed to reduce the overall runtime due to the high invocation cost.
2. RDC-SAT solver showed runtime improvement on both random 3-SAT and SAT competition benchmarks compared to the traditional D&C solvers. Particularly, the improvement on the SAT competition benchmark demonstrated the RDC-SAT’s effective generalization to industrial benchmarks, which many ML-based solvers could not achieve.
3. The authors also developed a tailored RL environment to handle the tree-like state transitions involved in the D&C framework.

**Weaknesses:**

Though it is an interesting idea for the ML-assisted solver, the evaluation results are slightly lower than what I expected. Out of the 305 instances from SAT competition 2023, only 2 more instances were solved by RDC-SAT than the most competitive baseline. Note that in the competition 2023, the state-of-the-art parallel SAT solver (PRS-parallel) solved 320 out of 400 instances, and I think the solved number over these 305 instances will still be considerably higher than RDC-SAT’s (135 out of 305).

**Questions:**

I am reluctant to give a high score considering the barely satisfactory evaluation results, especially if compared to the state-of-the-art parallel SAT solvers. Can you please elaborate on the specific scenarios or problem types where D&C solvers can outperform the portfolio solvers like PRS-parallel, supported by empirical evidence or citations? A good motivation for using D&C solvers may change my opinion. It would also be good to have a more comprehensive discussion of the trade-offs between D&C and portfolio approaches in the paper's introduction or related work sections.

It would be good to know PRS-parallel's runtime performance on these 305 instances. Could you please run the experiments and report the results?

---

> ### Author Response · Authors · 2024-11-20
>
> We appreciate the feedback provided by Reviewer 5BwV. We'd like to respond to your questions as follows:
> # RDC-SAT Performance and the Advantages of D&C Solvers(Weakness and Question)
>
> RDC-SAT does not yet match the performance of current SOTA Portfolio parallel solvers on the SAT Competition dataset. However, our experiments on this dataset primarily aim to evaluate RDC-SAT’s generalization capabilities rather than achieving SOTA performance. Additionally, the experimental setup used significantly fewer threads and less memory compared to the settings of the SAT Competition, which likely impacted the results.
>
> Despite this, D&C solvers demonstrate unique advantages in specific scenarios. For instance, we conducted a small-scale experiment on a dataset containing 100 randomly selected instances from the miter benchmarks, which are used to verify circuit equivalence. The timeout was set to 1000 seconds, with all methods using 16 threads and VSIDS as the branch heuristic for the Sequential Solver. Using the model trained in our work, the results are summarized below:
>
> | Method              | Solved | AVG Time(s)    | PAR2    |
> |---------------------|--------|--------|---------|
> | RDC–GPU–always (D&C)| **80** | **313.64** | **513.64** |
> | FLIPS (D&C)         | 76     | 333.66 | 573.66  |
> | PRS (Portfolio)     | 79     | 385.44 | 595.44  |
>
> This result shows that D&C solvers can outperform Portfolio solvers in specific domains. In this experiment, RDC-SAT solved the most instances and achieved the lowest PAR2, reducing PAR2 by 13.74% compared to the SOTA Portfolio solver PRS. The RDC-SAT model, trained on a random dataset, generalized effectively to the circuit equivalence dataset.
>
> D&C solvers are also inherently more scalable to large distributed environments, which Portfolio solvers struggle to handle. For example, Paracooba [1] demonstrated excellent scalability on clusters and cloud platforms, while Heule et al. [2] used a D&C solver to solve the Boolean Pythagorean Triples Problem on an 800-core distributed cluster in two days, showcasing the power of D&C solvers in tackling challenging problems with extensive distributed resources.
>
> In our paper, RDC-SAT primarily focuses on optimizing the splitting heuristic in D&C solvers, rather than creating a SOTA parallel solver. Beyond the splitting heuristic, there remains substantial room for improvement. For example, we are currently replacing Maple with CaDiCaL as the sequential solver and training the model on larger, more diverse datasets to further enhance the performance of D&C solvers, potentially narrowing the gap with SOTA Portfolio solvers.
>
> References:
>
> [1]Heisinger et al. Distributed cube and conquer with paracooba.
>
> [2]Heule et al.. Solving and verifying the boolean pythagorean triples problem via cube-and-conquer.

---

> > ### Comment · Reviewer_5BwV · 2024-11-22
> >
> > Thanks for your response. Please add the related works in your response to the paper to strengthen motivation in the future.
> >
> > Thanks for reporting the new results with PRS. However, the improvement is not significant even for the selected miter benchmarks. Overall, I still prefer a weak acceptance considering the gap with the SOTA solvers. Although you claim that there is substantial room for improvement, without factual results, I am concerned about the impact of this work given the current empirical evaluation.

---

> > > ### Author Response · Authors · 2024-11-25
> > >
> > > Thank you for your feedback. We have included the discussion of the related works in the revised version of the paper. Additionally, we have further clarified the intent and contributions of our work in the Global Official Comment.

---

### Author Response · Authors · 2024-11-25

We sincerely thank all the reviewers for their thoughtful discussions and valuable feedback. In the revised paper, we have incorporated changes based on the reviewers' suggestions, including improvements to the discussion of related work, the definition of RL actions, and explanations of features in the graph. Additionally, some reviewers raised concerns about our experimental design, such as the choice of datasets and comparisons with Portfolio solvers. Here, we aim to clarify the primary purpose and design of our experiments.

The primary goal of this paper is to explore the use of reinforcement learning (RL) to optimize the splitting heuristic in Divide-and-Conquer SAT solvers and improve their performance. To ensure that the observed performance improvements were entirely attributable to the splitting heuristic, we primarily compared RDC-SAT with other D&C solvers within the same framework. Comparisons with sequential solvers or portfolio solvers were not included, as such evaluations would not accurately isolate the impact of the splitting heuristic and fall beyond the scope of this work. Additionally, the experiments focused on evaluating the effectiveness and generalizability of the proposed method rather than developing a finely-tuned, universally high-performance model. Accordingly, we trained RDC-SAT on a small-scale random dataset and evaluated its effectiveness on more challenging random datasets as well as its generalizability on the SAT Competition dataset. The results demonstrated performance improvements of 25.08% and 14.95% on the two datasets, respectively. Despite being a simple evaluation model trained solely on random datasets, it demonstrated strong performance on application instances in the SAT Competition dataset, underscoring the effectiveness and generalizability of our approach.

In recent years, Portfolio solvers have outpaced D&C solvers in development. Elevating the performance of D&C solvers to higher levels requires innovative and impactful efforts. Our work advances D&C solvers by improving their splitting heuristics, enabling RDC-SAT to achieve SOTA performance within this category. Beyond the splitting heuristic, RDC-SAT still has significant room for improvement, as outlined in the "Further Improvements for RDC-SAT" section. Additionally, D&C solvers offer unique advantages in certain domains, such as scalability in distributed environments, which Portfolio solvers often struggle to achieve.

We hope this clarification helps reviewers better understand the intent and contributions of our work, as well as the purpose of our experimental design.

---

### Author Response · Authors · 2024-12-02

Dear Reviewers,

As the discussion period is nearing its end, we kindly ask reviewers to let us know if our responses have addressed your remaining concerns. If you have any additional questions or concerns, please let us know, and we would be glad to address them promptly. We also sincerely thank reviewers once again for your thoughtful suggestions and valuable feedback.

Best regards,

Authors of Submission 5781

---

### Meta-Review · Area_Chair_5vhJ · 2024-12-22

**Metareview:**

The paper proposes a new technique based on RL to improve Divide and Conquer SAT solvers.  The new technique utilizes a graph neural network (GNN) with RL to select the next variable to split on.  The strengths of this work are the demonstration that better splits are found and the overall solution time is reduced despite the overhead of GNNs and RL.  The weaknesses are the limited novelty of the approach and the slight empirical improvement in the class of divide and conquer solvers that are not state of the art for SAT.  While the approach is closely related to previous work that uses GNNs and RL for branching heuristics, this is the first application of GNN and RL for splitting heuristics.  The proposed approach still advances the classes of divide and conquer solvers.

**Additional Comments On Reviewer Discussion:**

The reviewers discussed the novelty of the approach and the significance of the work.  Strictly speaking, the work is novel since it is the first application of GNNs and RL to divide and conquer SAT solvers.  Hence there is novelty, but this novelty is limited since the idea of using GNNs with RL is common for branching heuristics.  The significance of the empirical results was also discussed.  Although the approach does not advance the state of the art, it is still advancing divide and conquer SAT solvers, though not by a large margin.

---

### Decision · Program_Chairs · 2025-01-22

Accept (Poster)